# WHAT IS THE COLOR OF *RED*? VISION–LANGUAGE MODELS PREFER TO READ RATHER THAN SEE

## ABSTRACT

A Visual Language Model (VLM) learns a joint understanding of image and text and generate texts based on this understanding. Yet when multiple visual cues within an image conflict—such as a written word and its ink color—we do not fully understand how the model decides which signal to prioritize. A classical psychological paradigm to study how conflicting cues affect decision is the Stroop test, where participants are shown words in incongruent ink colors (e.g., the word "red" written in blue) and are instructed to report the ink color rather than read the word. We adapt the Stroop paradigm to VLMs and study how conflicting cues in the written word or ink color influence model's behavior. Applying the Stroop test on a range of contrastive and generative VLMs suggests that the models we tested favor textual cues over color when text cue and color cue conflict. Analyzing the representation of the two cue types suggest that text cues in images are more salient than the color cues in CLIP's embedding space, where we conduct representational analyses. This difference in saliency also translates to different intervention success to steer the VLMs: we found that it is easier to steer the embedding to make the model favor text cues than color cues. Overall, using the Stroop test, our findings suggest that the evaluated models tend to "read" an image rather than to "see", and the saliency of the two cue types is reflected in their embedding space for the models and settings we study. We will release our dataset and code to support future research upon acceptance.

## 1 INTRODUCTION

*"Language disguises thought."* — *Ludwig Wittgenstein*

Vision–Language Models (VLMs) (Radford et al., 2021; Li et al., 2022; Liu et al., 2023) have become central tools for multimodal learning. These models align images and text within a shared representation space and, when given an image and a prompt, produce predictions conditioned on both modalities. As real-world images often contain ambiguous or competing signals, it is crucial to understand how these models resolve conflicts. When different signals suggest competing interpretations, which one guides the model's decision?

We look into the literature studying how humans respond to conflicting cues in psychology. A classical paradigm to study judgment under conflicting cues is the Stroop test (Stroop, 1935). In this task, participants view color words (e.g., RED, BLUE) presented in different ink colors and are instructed to name the ink color. When the word and color conflict, people exhibit slower responses and higher error rates, typically favoring the word over the ink color.

We adapt this paradigm to study VLMs' judgment when word and ink color conflict. To do so, we constructed a Stroop dataset in which each image contains a word (the text string) and an ink color (the ink color). Some word–ink pairs are congruent, while others are incongruent. We then evaluated a range of models—including contrastive models such as CLIP (Radford et al., 2021) and SigLIP (Zhai et al., 2023), as well as several generative VLMs (e.g., LLaVA) on this paradigm. Across the board, the models we tested show a clear preference for the word over the ink color.

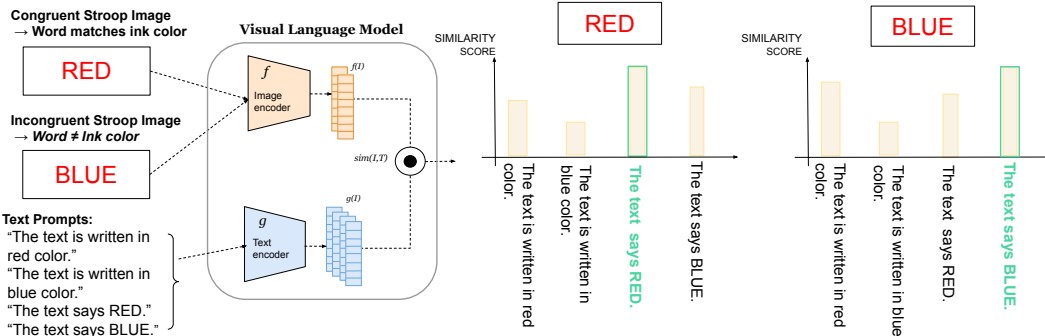

Figure 1: **Stroop-style evaluation of multimodal conflict.** Left: examples of *congruent* (word matches ink color) and *incongruent* (word conflicts with ink color) stimuli. Middle: a VLM maps the image through an image encoder $f(I)$ and each prompt through a text encoder $g(T)$; cosine similarity $sim(I,T)$ compares the resulting embeddings. Right: similarity scores for the same image under two prompt families: *word-oriented* prompts ("The text says BLUE") and *ink-oriented* prompts ("The text is written in red"). In incongruent cases, the higher bar typically corresponds to the word-oriented prompt, indicating that the model aligns more with the written word than with the ink color.

To study representations driving behavior, we analyzed CLIP's embedding space and found that word shape is represented more strongly than ink color. We then steered the preference of CLIP by perturbing neural subpopulations encoding concepts for words and ink colors. We observed that the representation saliency of a concept is mirrored in steerability: embedding representations encoding *word concepts are clearer, more separable, and reliably steerable*, whereas neural representations encoding *ink-color directions are weaker and more entangled.* As a consequence, it is easier to shift the VLM's representation encoding word than ink color.

Taken together, our findings show that under conflicting cues — as in the Stroop paradigm — the evaluated models tend to "read" the word rather than "see" the color. This preference is reflected in their representations and can be manipulated through steering. More broadly, our study illustrates a case of cue domination in VLMs revealed by psychology-inspired paradigms.

## 2 RELATED WORK

Vision-Language Models (VLMs) map images and text into a shared semantic space and underpin a wide range of applications, including captioning (Xu et al., 2015; Hossain et al., 2019), retrieval (Faghri et al., 2018), and visual question answering (Antol et al., 2015; Alayrac et al., 2022; Tsimpoukelli et al., 2021). Architecturally, two main families dominate the literature: dual encoders that align image and text embeddings with contrastive learning (exemplified by CLIP (Radford et al., 2021)); and encoder–decoder pipelines that condition generation on visual features (e.g., BLIP-2 (Li et al., 2022), Flamingo (Alayrac et al., 2022), LLaVA (Liu et al., 2023), Qwen2-VL (Wang et al., 2024)). While these systems achieve strong zero-shot performance, less is known about how they resolve *multimodal conflict*—cases in which visual and linguistic signals pull in opposite directions.

Beyond multimodal systems, even purely visual CNNs exhibit systematic representational biases—for example, favoring local texture statistics over global shape information (Geirhos et al., 2022). Prior work has further shown that increasing shape bias improves both robustness and generalization, suggesting that analogous strategies might help counteract word bias in VLMs. Alongside behavioral evaluations, the interpretability literature offers tools for inspecting representation structure. Cosine-based similarity diagnostics, Representational Dissimilarity Matrices (RDMs) (Kriegeskorte et al., 2008), and low-dimensional visualizations are commonly used. More recent latent steering methods move activations along concept directions, either via "chunks" learned from data (Wu et al., 2025) or monosemantic units (Pach et al., 2025; Goh et al., 2021). These techniques mainly emphasize *word* concepts; systematic comparisons between *word* and *ink-color* directions remain scarce. Complementary analyses such as VLM-Lens (Sheta et al., 2025), visual-illusion probes (Zhang et al.,

2023), and multimodal cognition benchmarks (Buschoff et al., 2024) highlight the broader need for fine-grained evaluation of how conflicting cues are represented.

# 3 ADAPTING STROOP TASK FOR VLMs

We constructed a Stroop-style dataset of images following the classic paradigm (Stroop, 1935). Each image displays a *word* rendered in an *ink color*. Images are divided into two categories: *congruent*, where the word and ink color match, and *incongruent*, where the word and ink color conflict. The dataset contains 100 images balanced across ten basic colors (red, blue, green, yellow, orange, pink, purple, black, brown, and gray). Each color appears both as a written word and as an ink color across stimuli, ensuring full coverage of congruent and incongruent combinations. For all models, we evaluate two types of cues: a *word-oriented* prompt set ("The text says RED") and an *ink-oriented* prompt set ("The text is written in red color"). Each Stroop image is evaluated in two separate forward passes, one per prompt family. In each pass, the model selects the highest-scoring prompt (contrastive models) or produces an output mapped to one of the ten color classes (generative models), and accuracy is computed relative to the corresponding ground-truth attribute (word or ink). This unified protocol governs all behavioral analyses in the paper; model-specific scoring details (contrastive vs. generative) are provided in Sec. 4 and Sec. 5. This setup isolates a single, well-defined conflict type—word vs. ink color—allowing us to study a narrowly scoped form of multimodal cue competition.

# 4 BEHAVIORAL ANALYSIS FOR CLIP AND SIGLIP-2

We begin with CLIP (Radford et al., 2021), the canonical contrastive Vision–Language Model, which we use as a baseline for Stroop-style evaluation and still used widely in recent research (Koleilat et al., 2025; Dong et al., 2025; Liu et al., 2024; Hossain & Imteaj, 2024).

Similarity between an image and a text prompt is computed using cosine similarity: $s(I,T) = \frac{f(I)\cdot g(T)}{\|f(I)\|\,\|g(T)\|}$ where $I$ is the input image, $T$ the text prompt, $f(\cdot)$ the image encoder, and $g(\cdot)$ the text encoder. On congruent images, both evaluations achieve ceiling performance, as expected. On incongruent images, the word-oriented evaluation reaches **97.8%** accuracy with respect to the ground-truth word, while the ink-oriented evaluation dramatically drops to about **20%** accuracy with respect to the ground-truth ink color (Figure 2).

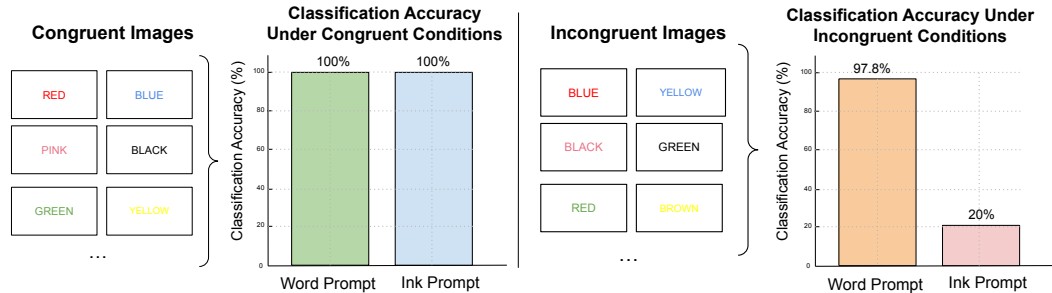

Figure 2: **CLIP behavioral results.** Left: examples of *congruent* (word matches ink color) stimuli, where both evaluations reach ceiling performance. Right: results for incongruent conditions, showing that word- and ink-oriented evaluations diverge sharply. Accuracies are computed independently in separate forward passes and are not complementary.

We next examine a variant of CLIP: SigLIP-2 (Tschannen et al., 2025) on the same paradigm. SigLIP-2 replaces CLIP's softmax contrastive loss with a sigmoid loss and adds multilingual and variable-resolution support. SigLIP-2 outputs image–text logits, which are passed through a sigmoid and normalized across the prompt set; the predicted label is the prompt with the highest resulting probability.

| Model | Congruent | Incongruent | |
|---|---|---|---|
| | Word = Ink (%) | Word Accuracy (%) | Ink Accuracy (%) |
| CLIP | 100.0 | 97.8 | 20.0 |
| SigLIP-2 | 100.0 | 100.0 | 5.6 |

Table 1: **Results for CLIP and SigLIP-2 on Stroop stimuli.** Accuracy is defined as the percentage of images where the higher-scoring prompt corresponds to the word (*word accuracy*) or to the ink (*ink accuracy*). The "Word = Ink" column reports performance on congruent cases, where the word and the ink are the same. In incongruent cases, the word-oriented evaluation yields very high accuracy for both models, whereas the ink-oriented evaluation yields much lower accuracy, particularly for SigLIP-2.

Using this criterion, we evaluated SigLIP-2 on the Stroop images and observed a similar behavior to CLIP. While SigLIP achieves ceiling performance for both word prompts and color prompts, the word-oriented evaluation on SigLIP-2 reaches **100%** accuracy with respect to the ground-truth word, while the ink-oriented evaluation dramatically drops to about **5–6%** accuracy with respect to the ground-truth ink color. Both CLIP and SigLIP-2 assign higher similarity to word-oriented prompts but have trouble finding the correct color-oriented prompts in incongruent Stroop images. To ensure that the observed word bias was not specific to the initial dataset, we tested whether this observation persisted under a set of controlled variations of the Stroop tasks. Specifically, we varied font size (48–108 pt), font weight (light, normal, bold, narrow), contrast (high, medium, low, and a *same* condition where letters blend with the background), and pseudowords that preserve visual form while removing semantic content. In the visual manipulation experiments for these extended Stroop tasks, we used a single template prompt, "The text is written in {ink} color," where {ink} was instantiated with 10 candidate terms (red, blue, green, . . . , black). For each Stroop image (congruent and incongruent), the model received these 10 prompts, and we recorded the highest-scoring one as its prediction. If the predicted ink matched the true font color of the stimulus, we counted it as *ink accuracy*; if it matched the written word, this indicated a *word bias*; and if it matched neither (e.g., a background color), it was classified as *other*.

Evaluating CLIP on the extended set of Stroop tasks -varying contrast, font size, font weight, and pseudowords- confirms the persistence of this preference. CLIP favors the word whenever it is legible. The preference gap narrows only when the word becomes unreadable (e.g., 108 pt overflow, zero contrast), at which point the model shifts toward the ink. As a control, in pseudoword conditions, the model's preference also flips to the ink color, as semantic content is absent. Across these manipulations, the dominance of text bias in CLIP is striking: for most legible conditions, word-based predictions remain in the 85–97% range, while ink accuracy rarely exceeds 20–30%. Only when legibility is strongly reduced -such as at 108 pt overflow, in the *same* contrast condition, or with pseudowords- does the model shift toward ink-based predictions. These results confirm that the strong word bias observed in the base Stroop task is not incidental, but persists robustly across visual manipulations. Beyond the controlled manipulations, we also tested CLIP on a much larger Stroop-style corpus of 23,338 images with diverse backgrounds, tones, and textures. This large-scale setting removes template-based shortcuts and requires genuine color recognition. The pattern remained unchanged: word accuracy stayed near-perfect (99.5%), while ink recognition remained low (15.9%), confirming that the word bias persists even under substantial visual diversity.

For **SigLIP-2**, we observe an even stronger tendency than CLIP. We evaluated the model separately with two sets of prompts: ink-oriented ("the text is written in {ink} color") and word-oriented ("the text says {word}"). For each image, the model scored all 10 candidates within a set, and the highest-scoring prompt was taken as the outcome. Accuracy was then measured relative to the ground truth: selecting the ink color (*ink accuracy*), the written word (*word accuracy*), or neither (*other*).

The word-bias persists in SigLIP-2 across variations of Stroop images in font sizes (48–108 pt) and font weights (light, normal, bold, narrow): the model assigns the highest similarity to the word almost always, and to the ink color only around 15–16% of the time. Contrast manipulations reduce the gap slightly: at medium and same-contrast levels, the model sometimes produces near-ties, but as far as the letters in the image remain legible, the model assigns the highest similarity to the prompt that contains the word. Finally, in pseudoword trials, where semantic content is removed, SigLIP-2 shifts the highest similarity assignment toward the ink color ∼67% of cases. Full results are provided

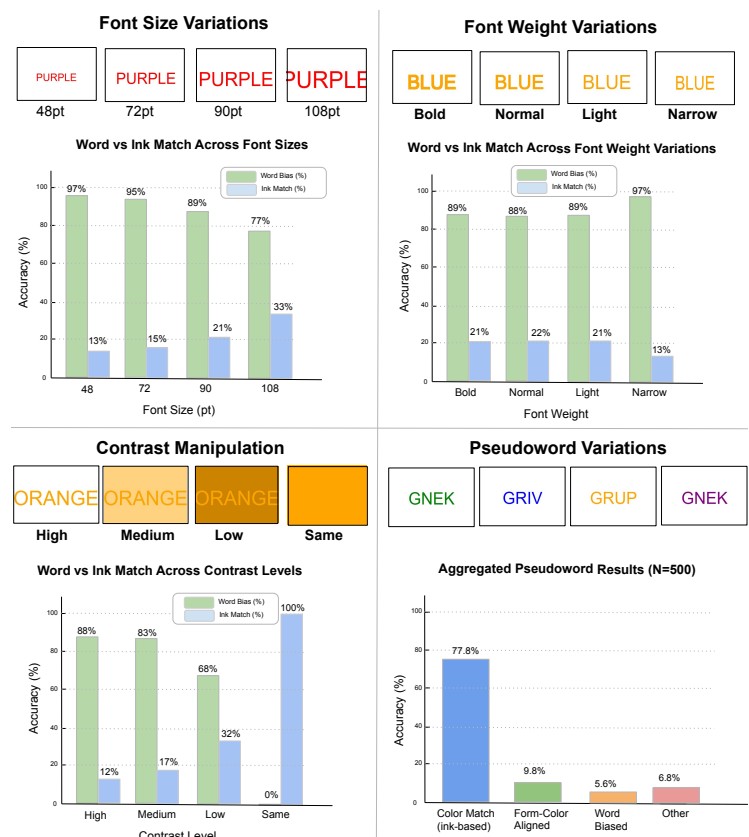

Figure 3: **Legibility controls cue preference** Top: examples from the four manipulation families (font size, font weight, contrast, pseudowords). Bottom: per-condition outcomes aggregated across colors. Prompts followed the template "The word is written in {ink} color" with 10 candidate colors. For each image, the model's prediction was defined by the highest-scoring prompt: selecting the true font color (**ink accuracy**), the written word (**word bias**), or neither (**other**). On the base Stroop dataset, CLIP shows only ∼13% ink accuracy versus ∼97% text bias, confirming a strong tendency to read rather than see. Under pseudoword conditions, where semantic content is removed, responses flip toward the ink.

in Appendix F. Overall, as long as the word remains legible, word preference persists for both CLIP and SigLIP-2 despite variations in font size, weigh,t and contrast.

## 5 BEHAVIORAL ANALYSIS OF GENERATIVE VLMs

While CLIP computes similarity scores between image and prompt, most contemporary Vision–Language Models are *generative*: given an image and an instruction, they produce a free-form textual answer. To test whether the Stroop pattern persists in this setting, we evaluate six open-source generative VLMs—BLIP-2 (FlanT5-XL), InstructBLIP (Vicuna-7B), Kosmos-2 (1B), LLaVA (Vicuna-7B v1.6), GIT (B/16), and Qwen2-VL-7B-Instruct—using a single English instruction designed to target the ink color: *"What color is the word in this image?"* In practice, answers vary: some models name the ink color, some repeat the written word, some explicitly mention both (e.g., "The word BLUE is written in red"), and some drift off-target. We then map each output into one of four categories: *Ink Match* when the ink color is correctly named, *Word Match* when the written word is repeated, *Both* when both are explicitly mentioned, and *Neither* otherwise. Color synonyms are normalized (e.g., *scarlet→red*), and multi-color responses are treated as *Both*. For compactness we also report a three-way split grouping *Both* under Ink Match. Exact label mappings are detailed in Appendix E. All models are evaluated on the same 100-image Stroop set (10 congruent, 90 incongruent) used for CLIP. Since the instruction explicitly asks for the ink color, we report *Ink Match* as the primary accuracy measure, while *Word Match*, *Both*, and *Neither* provide complementary breakdowns.

Across a range of generative VLMs, we observed a persistent word bias. While all models perform at their ceiling for congruent images, the generative VLM exhibits different levels of preference for incongruent images. Despite being explicitly instructed to report the ink color, most models generate a response containing the written word. Kosmos-2 showed the strongest word bias, almost never producing the ink color; BLIP-2 and InstructBLIP fell in between, with occasional ink-aligned answers but a clear bias toward reporting the word; and GIT was unstable, with a sizable fraction of response that neither contains the word nor the ink color *Neither*. LLaVA stood out as the only model that actually reported the ink color more frequently than the word, but ink color occurs in the prompt in roughly half of the generated text (54.4%), while still in 45.6% of the cases the network answers the prompt incorrectly with the word.

To test whether scale and instruction-tuning can alleviate this bias, we additionally evaluated **Qwen2-VL-7B-Instruct** (Wang et al., 2024), a recent large-scale, instruction-tuned vision–language model with 7B parameters that extends Qwen2-VL by incorporating stronger multimodal pretraining and alignment. As expected, it answers the ink-color question perfectly for congruent stimuli. On incongruent inputs, however, success depended heavily on the exact prompt wording. Specifically, with the longer instruction (*"You will see a single English word rendered in a colored ink. Ignore the written word and answer ONLY the ink color as one lowercase color name."*) the model achieved 31.1% ink accuracy, whereas with the shorter variant (*"What is the ink color of the text in the image? Answer with one lowercase color word only."*) performance increased to 60.0% (Full experimental setup and extended results for Qwen2-VL are provided in Appendix H). This sensitivity shows that even advanced instruction-tuned models still exhibit Stroop-style word bias, and their apparent robustness may vary substantially with phrasing.

In sum, across the Stroop setting, most generative VLMs tend to repeat the written word in incongruent cases, although the strength of this tendency varies substantially across architectures and depends on prompt phrasing.

| Model | Congruent | | | Incongruent | | |
|---|---|---|---|---|---|---|
| | Ink Match (%) | Word Match (%) | Neither (%) | Ink Match (%) | Word Match (%) | Neither (%) |
| BLIP-2 | 100.0 | 100.0 | 0.0 | 36.7 | **90.0** | 4.4 |
| InstructBLIP | 100.0 | 100.0 | 0.0 | 28.9 | **67.8** | 3.3 |
| Kosmos-2 | 100.0 | 100.0 | 0.0 | 4.4 | **95.6** | 0.0 |
| GIT | 80.0 | 80.0 | 20.0 | 12.2 | **68.9** | 18.9 |
| LLaVA | 90.0 | 90.0 | 10.0 | **54.4** | 45.6 | 0.0 |
| Qwen2-VL-7B | 100.0 | 100.0 | 0.0 | 31.1 (main) | **68.9** | 0.0 |
| | | | | **60.0** (alt) | 40.0 | 0.0 |

Table 2: Behavioral results for six generative VLMs on 10 congruent and 90 incongruent Stroop images, evaluated under the instruction *"What color is the word in this image?"*. Percentages show the fraction of trials labeled as Ink Match, Word Match (bold), or Neither. Cases where the output mentions both the word and the ink color ("Both") are counted under Ink Match, since the generated text still includes the correct ink information. Qwen2-VL-7B-Instruct is shown under two prompt conditions.

# 6 Studying the Representation of Word and Ink Color in Embedding Space

We next ask whether the observed word-bias also arises from the model's internal representation space. To examine this, we focus on **CLIP**, whose contrastive encoder architecture exposes image–text embeddings directly and allows systematic probing of how word and color information are organized. Specifically, we first analyze the **embedding representation** of CLIP to examine whether word and ink information are encoded with equal saliency by computing the Representational Dissimilarity Matrices (RDMs). RDM quantifies how dissimilar different stimuli are in embedding space (Moerel & Grootswagers, 2025). Concretely, given embeddings $e_i$ and $e_j$ for two stimuli, each entry of the RDM is defined as their cosine dissimilarity: $\text{RDM}(i, j) = 1 - \frac{e_i \cdot e_j}{\|e_i\| \|e_j\|}$. Higher values indicate that the model represents the two stimuli more differently in its embedding. We constructed RDMs for three input variants: *ink-only* (solid color backgrounds without text), *word-only* (grayscale words without color), and *word+ink* (colored words combining both cues). **RDM(Word+Ink)** refers to the full pairwise dissimilarity matrix among embeddings of Stroop images containing both word

and ink color, while **RDM(Ink-only)** and **RDM(Word-only)** provide corresponding baselines where either modality is present.

To isolate the incremental contribution of each modality, we computed differential RDMs by subtraction: $\Delta$Word = RDM(Word+Ink) − RDM(Ink-only), $\Delta$Ink = RDM(Word+Ink) − RDM(Word-only). Importantly, we do not interpret these differential RDMs as a causal decomposition of the representation. All embeddings are $\ell_2$–normalized before cosine dissimilarity is computed, so subtraction simply provides a comparable scale across matrices. The resulting differences should be read as an *approximate indication* of how the geometry changes when moving from a cue-isolated baseline to the full (word+ink) stimulus, rather than as the effect of a single modality in isolation. This framing avoids overinterpretation while still highlighting which cue contributes more strongly to the observed representational shifts.

Shown in Figure 4, $\Delta$Word panel shows stronger and more localized red regions, indicating that adding the word produces larger and sharper representational shifts. By contrast, $\Delta$Ink produces weaker and more diffuse changes, suggesting that ink contributes less the separation of embedding space than the written word.

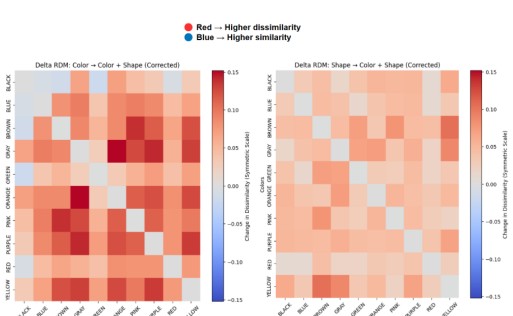

(a) **Differential RDMs for CLIP.** Left: $\Delta$Word (adding the written word on top of ink). Right: $\Delta$Ink (adding ink color on top of word).

(b) **UMAP projection of CLIP embeddings.** Text prompts cluster tightly by semantics, while image embeddings distribute more broadly yet still group by the written *word* rather than by ink. Colored points denote specific ink classes; connecting lines link the word- and ink-oriented variants of the same class.

Figure 4: **RDM and UMAP analyses for CLIP.** Left: Differential RDMs showing modality-specific contributions. Right: UMAP embedding view highlighting word-dominance over ink.

To visualize this embedding space, we additionally project both Stroop images and text prompts into two dimensions using UMAP (McInnes et al., 2018). For consistency, we used a unified prompt format ("The text says RED written in red color"), which explicitly encodes both cues. The resulting projection in Figure 4 shows that text prompts cluster compactly by semantics, whereas image embeddings distribute more broadly but still align according to the written *word* (see Appendix C for extended discussion). Together, the RDM and UMAP analyses suggest that CLIP's word preference in behavior is reflected in distinct representational saliencies for word and ink color in the embedding space (see Appendix C).

## 7 STEERING REPRESENTATIONS

Building on this representational evidence, we then test whether we can steer the image embeddings towards concept-encoding directions.

We adapt the population averaging method in Wu et al. (2025) to extract concept-encoding embedding dimensions. To do this, for each concept (e.g., "red"), we collected embeddings from all images containing that concept. For *ink-color chunks* (e.g., red), this included all images written in red ink regardless of the word; for *word chunks* (e.g., "RED"), this included all images containing the word "RED" regardless of ink color. From these embeddings, we identified a **subpopulation of**

**stable dimensions** that responded consistently to the concept. Within this subspace, we computed the average embeddings for the source ($\mu_{\Omega(\text{src})}$) and target ($\mu_{\Omega(\text{tgt})}$) concepts, where $\Omega$ denotes the identified concept encoding dimensions. Given an embedding $E$ of a *congruent* sample (e.g., "RED" in red), we restrict $E$ to the same subspace ($E_\Omega$) and apply an intervention as $E'_\Omega = E_\Omega - \mu_{\Omega(\text{src})} + \mu_{\Omega(\text{tgt})}$. Intuitively, it removes the contribution of the source concept and replaces it with that of the target within the subspace where the concept is consistently represented.

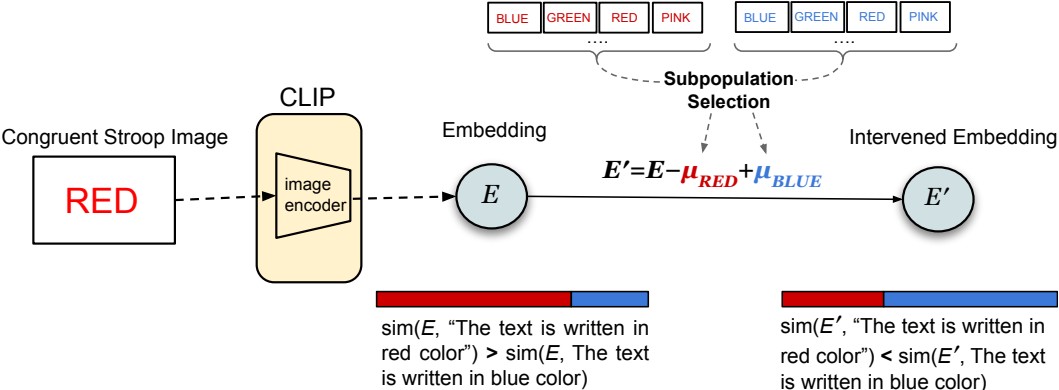

Figure 5: **Subpopulation-based intervention pipeline.** A Stroop stimulus (e.g., the word "RED" in red ink) is encoded by CLIP into an embedding $E$. Interventions are applied only on **congruent inputs**, ensuring that each embedding is aligned with a single concept before editing. For each concept (e.g., red, blue), embeddings are aggregated across all relevant examples (for colors, all items written in that ink color; for words, all items containing that word). Per-dimension variance is then computed, and dimensions are clustered by stability; the low-variance subset defines the stable subspace. Source ($\mu_{\text{src}}$) and target ($\mu_{\text{tgt}}$) averages are computed within this subspace, and the intervention is applied as $E' = E - \mu_{\text{src}} + \mu_{\text{tgt}}$. Finally, cosine similarity with source and target prompts is re-evaluated. The circle highlights the intervened embedding $E'$ in representation space.

We consider three types of interventions: *ink-color steering* (e.g., red → blue), *word steering* (e.g., "RED" → "BLUE"), and a *combined* intervention that applies both shifts. All interventions are evaluated on **congruent** examples, where the word and ink color match (e.g., "RED" in red). Here, the **source** refers to the original concept of the image (e.g., red) and its corresponding source prompt embedding (e.g., "The text says RED"). The **target** denotes the intended concept after intervention (e.g., blue), evaluated using its target prompt embedding. Note that the chunk vectors are only used to modify the image embedding; evaluation is always performed against prompt embeddings. For each edited embedding, we then measure the change in cosine similarity between the target and source prompts: $\Delta = \text{sim}(E', \text{target}) - \text{sim}(E', \text{source})$, and count an intervention as *successful* if $\Delta > 0$. For example, suppose we start with a congruent image of the word RED written in red ink. In a *color steering* intervention, we shift the embedding toward the concept blue. We then compare the modified embedding to the prompt "The text is written in blue color" (target) versus "The text is written in red color" (source). In a *word steering* case, we instead steer toward the word BLUE and evaluate against "The text says BLUE" (target) versus "The text says RED" (source). Finally, for a *combined* intervention, we simultaneously shift along both dimensions, testing against "The text says BLUE, written in blue color" (target) versus "The text says RED, written in red color" (source).

Word chunks achieve **100% success**, with average similarity shifts of $+0.0934 (\pm 0.0214)$. Combined edits also succeed in **100%** of cases $(+0.1172 (\pm 0.0215))$. Ink-color chunks perform substantially worse, reaching only **36.67%** success and a near-zero average shift $(-0.0017 \pm 0.0166)$. Diagnostics clarify why: word encoding vectors have much larger $\ell_2$ norms (mean **6.36**) and are more distinct, while ink-color encoding vectors are shorter (mean **2.99**) and highly collinear (average cosine $\approx 0.79$ after subpopulation filtering). Thus, within CLIP, color edits rarely succeed in steering the embedding reliably.

Figure 6 shows the detailed heatmaps of $\Delta$ values for each source–target pair under Ink, Word, and Combined interventions. Positive $\Delta$ (red) indicates successful steering toward the intended target. While Word and Combined interventions are easier to steer and achieve strong and consistent

positive shifts across all pairs, Ink-Color interventions are harder to steer and steering shows weak or diffuse effects, often hovering around zero. Implementation details, parameter sweeps, and additional diagnostics are provided in Appendix D.

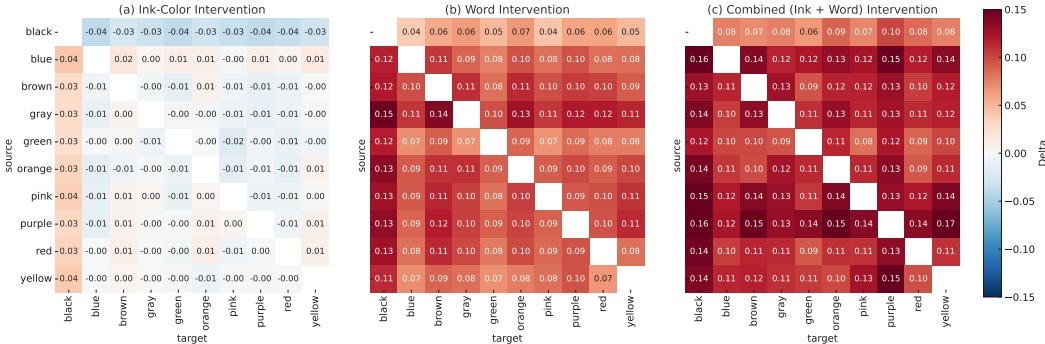

Figure 6: Per-concept intervention effects in CLIP's embedding space. (a) Ink-Color intervention, (b) Word intervention, and (c) Combined (Ink + Word). Heatmaps display $\Delta = \text{sim}(E', \text{target}) - \text{sim}(E', \text{source})$ for each source–target pair. White denotes $\Delta \approx 0$, blue denotes negative shifts, and red positive shifts. Successful steering corresponds to $\Delta > 0$, i.e., target similarity exceeding source similarity. Word and Combined interventions yield consistently high positive $\Delta$ values (compact, modular steering), whereas Ink-Color interventions remain weak and inconsistent (low or near-zero $\Delta$).

Table 3: Success rate of subpopulation-based interventions. Success is defined as $\Delta > 0$, i.e., target similarity exceeding source similarity after intervention.

| Intervention Type | Success (%) | Mean $\Delta$ (± Std) |
|---|---|---|
| Ink-Color | 36.67 | $-0.0017 \pm 0.0166$ |
| Word | 100.0 | $+0.0934 \pm 0.0214$ |
| Combined | 100.0 | $+0.1172 \pm 0.0215$ |

Beyond the CLIP analysis, we applied the same layer-wise steering method to two generative VLMs, Qwen2-VL-7B and LLaVA-1.6-7B, evaluating how color-, word-, and combined-direction edits propagate across depth. At the final layer of each model, steering remained effective but differed in strength: Qwen2-VL achieved perfect color and combined steering (both 100%) and near-perfect word steering (97.8%), whereas LLaVA showed moderately lower rates, with 94.4% for color steering, 73.3% for word steering, and 96.7% for combined steering. Full layer-wise trajectories and matrices are reported in Appendix I.

## 8 GENERALIZING MULTIMODAL CONFLICT BEYOND TEXT AND COLOR

To test whether Stroop-style conflicts also arise in more naturalistic settings, we generated a set of realistic multimodal conflict images using the FLUX.1 model. These include contradictory cases such as a green "STOP" sign or a red Wi-Fi icon labeled "CONNECTED". For each image, we paired a *word prompt* (describing the written content) with a *color prompt* (describing the dominant visual cue), and measured CLIP's similarity to determine whether it favored the textual or visual signal. Across these stimuli, CLIP alternated between word-based and color-based decisions depending on cue saliency: text-heavy signage induced word dominance, whereas icon-like or strongly chromatic regions elicited color dominance. Two examples are shown in Table 4; the full set with similarity scores and interpretations is provided in Appendix J.

## 9 CONCLUSION

In this work, we adapt the Stroop paradigm and systematically test Vision–Language Models' behavior in the presence of conflicting cues between the *word* (the printed string) and the *ink*

Table 4: **Illustrative FLUX-generated conflict cases.** CLIP's decision reflects either the textual or visual cue, depending on saliency.

| Image | Visual Description | word_sim | color_sim | Decision |
|---|---|---|---|---|
|  | "STOP" text in green sign | 0.17 | 0.83 | COLOR |
|  | "EXIT" (red sign) | 0.99 | 0.001 | WORD |

*color*. We observed a consistent behavioral bias across contrastive encoders (CLIP, SigLIP) and generative VLMs (BLIP-2, InstructBLIP, Kosmos-2, LLaVA, GIT, Qwen2-VL): when word and ink color disagree, models overwhelmingly align their preferences with the word, similar to humans. This tendency persists despite varying font size, font width, and contrast. Going beyond behavioral observations, we analyzed CLIP's embedding space and found that neural subpopulation vectors encoding words are long and modular, whereas those encoding ink-color are short and collinear. This representation saliency is reflected in steerability: steering embedding space towards word directions consistently flipped the network's preference, while steering towards ink-color direction is less effective. This suggests that word over color dominance is represented in the embedding space. Together, by adapting a classic psychological paradigm to VLMs, our study shows that models prioritize word over color cues when they conflict, a bias rooted in unequal embedding representations. Our systematic study on multi-cue conflict in the Stroop test provides a foundation for future understanding of cue representation in the most general setting.

## 10    DISCUSSION AND LIMITATIONS

Our study has limitations. The dataset, while faithful to the Stroop paradigm, is limited to ten basic colors and simple stimuli, and thus may not capture the broader range of cue conflicts; due to limited computational resources, embedding-level interventions were conducted only in CLIP. And our assessment about generative VLM (WordMatch/InkMatch/Both/Neither) may miss edge cases, for instance, when models produce descriptive sentences mixing both attributes. Additionally, chunk extraction for ink color likely underestimates steerability if color is more distributed than our centroids capture, and SigLIP's rigidity under legibility manipulations suggests model-specific sensitivities that we did not probe exhaustively. Future work may extend latent interventions beyond CLIP to recent VLMs (e.g., LLaVA-1.6, InternVL-2.5, Qwen-2.5-VL, SigLIP-2) and to multi-image or video inputs, and move from synthetic to naturalistic conflicts (textures, materials, layered graphics). and develop stronger ink/color representations in the embedding space, for example, by improving chunking methods, using training data that emphasizes color independently of semantics, or exploring architectural approaches that explicitly decouple reading from seeing. Prior work has documented broader forms of textual dominance (Menon et al., 2022; Pezeshkpour et al., 2025; Deng et al., 2025) and difficulty integrating conflicting visual–textual cues (Jia et al., 2025), as well as color-processing weaknesses in contrastive encoders (Arias et al., 2024). However, these analyses do not isolate the specific, tightly-controlled word–vs–color conflict we study, nor do they connect behavioral asymmetry to representational geometry or steerability. Our Stroop-style paradigm addresses this gap by providing a mechanistic explanation of why word cues dominate. Furthermore, future work may examine scaling effects within the same model family (e.g., Qwen2-VL 7B → 13B) to test whether the word–color asymmetry weakens with capacity. A complementary direction is to analyze token-probability dynamics, which may reveal Stroop-like slowdowns in model confidence that are not captured by final accuracy alone.

ETHICS STATEMENT

This work investigates modality bias in Vision–Language Models through Stroop-style conflicts. Large Language Models (LLMs) were used only to aid and polish the writing; no ideas, analyses, or experiments were generated by them. We believe this disclosure is important for transparency.

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

*Appendices* Dataset and prompt details are in Appendix A (A); full behavioral tables and manipulation results in Appendix B (B); embedding-space diagnostics (RDM/UMAP) in Appendix C (C); latent intervention methods and analysis in Appendix D (D); and word/ink-color label mapping with post-processing rules in Appendix E (E).

# A  STROOP DATASET AND PROMPT DESIGN

To investigate how Vision–Language Models (VLMs) handle multimodal conflicts, we constructed a synthetic Stroop-style dataset used across all experiments in the paper.

## A.1  DATASET OVERVIEW

The dataset consists of **100 Stroop stimuli** generated programmatically for perfect balance. Each image contains a single uppercase color word rendered in a specific ink color.

- **Color vocabulary (10 classes):** *red, blue, green, yellow, orange, purple, brown, pink, gray, black*.
- **Stimulus types:** **10** congruent (word==ink) and **90** incongruent (word≠ink).
- **Image design:** white background, centralized positioning, uniform font (90 pt).
- **Resolution & format:** 256×256 px PNG (sRGB, embedded ICC profile).
- **Filenames:** `<WORD>_<INK>.png` (e.g., `BLUE_RED.png`).

## A.2  DATASET BALANCE

Each of the ten words appears exactly once in each of the ten ink colors, yielding a perfectly balanced set of **100** stimuli in total (**10 congruent**, **90 incongruent**). Balance therefore holds by both word and ink: every color word is rendered once in its matching ink and nine times in non-matching inks, and the same symmetry holds when counted by ink color.

## A.3  COLOR SPECIFICATION

We fix sRGB hex values to avoid palette drift and to ease reproduction.

| Color | red | blue | green | yellow | orange | purple | brown | pink | gray | black |
|---|---|---|---|---|---|---|---|---|---|---|
| Hex | fc0404 | 0c0cfc | 048404 | fcfc17 | fca404 | 8c178c | 8c4414 | fcc4cc | 848484 | 000000 |

Table 5: Fixed sRGB hex codes for ink colors.

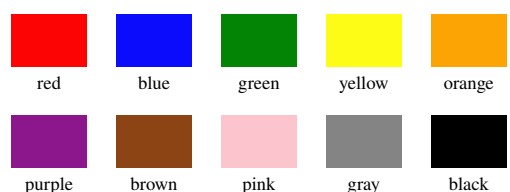

red  blue  green  yellow  orange
purple  brown  pink  gray  black

Figure 7: sRGB ink swatches used in the dataset (correspond to Table 5).

Each model was evaluated on all 100 Stroop images (10 congruent, 90 incongruent). For visual manipulation experiments (font size, weight, contrast, pseudowords), we generated an additional 400–500 samples per manipulation, resulting in a few thousand evaluation datapoints overall.

## A.4  EVALUATION SCALE (DATAPOINTS)

## A.5  PROMPT DESIGN

We use two prompt families for CLIP and a single standardized question for generative VLMs.

| Experiment type | #stimuli | #models | Total evaluations |
|---|---|---|---|
| Base Stroop (cong+incong) | 100 | 7 | 700 |
| Font size (4×100) | 400 | 2 (CLIP+SigLIP) | 800 |
| Font weight (4×100) | 400 | 2 | 800 |
| Contrast (4×100) | 400 | 2 | 800 |
| Pseudowords | 500 | 2 | 1000 |
| Qwen2-VL (2 prompts) | 100 | 1 | 200 |
| **Total** | – | – | ~4,300 |

Table 6: Scale of evaluation datapoints across experiments (updated with Qwen2-VL).

- **Ink-oriented (visual):** "The text is written in red color."
- **Word-oriented (semantic):** "The text says RED."

**Generative VLM query:**

*"What color is the word in this image?"*

For Qwen2-VL-7B-Instruct we used two prompt variants; see App. H.

## A.6 VISUAL MANIPULATIONS

We additionally probe legibility and perceptual robustness via controlled perturbations.

| Manipulation | Description | Purpose |
|---|---|---|
| Font size | 48, 72, 90, 108 pt | Visual salience / overflow check |
| Font weight | Bold, Normal, Light, Narrow | Readability vs. ink reliance |
| Tone variation | 10 brightness/saturation levels | Perceptual robustness |
| Contrast | High / Medium / Low / Same (text color equals background) | Visibility thresholding |
| Pseudowords | Word replaced by nonwords (e.g., GNEK, GRIV, GRUP, ...) | Remove lexical semantics |

Table 7: Visual manipulations applied to Stroop stimuli.

## A.7 EXAMPLE STIMULI

## B FULL BEHAVIORAL RESULTS

This appendix expands the CLIP analyses with precise counting rules, per-condition numbers, and the effects of visual manipulations.

### B.1 CLIP BEHAVIORAL RESULTS

**B.1.1 Counting rule** Unless stated otherwise, a prediction is taken via a *winner-takes-all* cosine score between the *ink-oriented* and *word-oriented* prompt families Section 4. We then label the outcome as **Ink Match** (predicted ink color), **Word Match** (predicted written word), or **Neither**.

| Condition | Ink Match | Word Match | Neither |
|---|---|---|---|
| Congruent (n=10) | 0 | 10 | 0 |
| Incongruent (n=90) | 9 | 76 | 5 |

Table 8: Winner-takes-all counts under congruent vs. incongruent stimuli.

**B.1.2 Family-specific accuracies (for comparability with the main text)** For completeness, we also report accuracies *within* each prompt family, i.e., scoring the image against the color prompts only, or against the text prompts only. These are the numbers cited in the main paper.

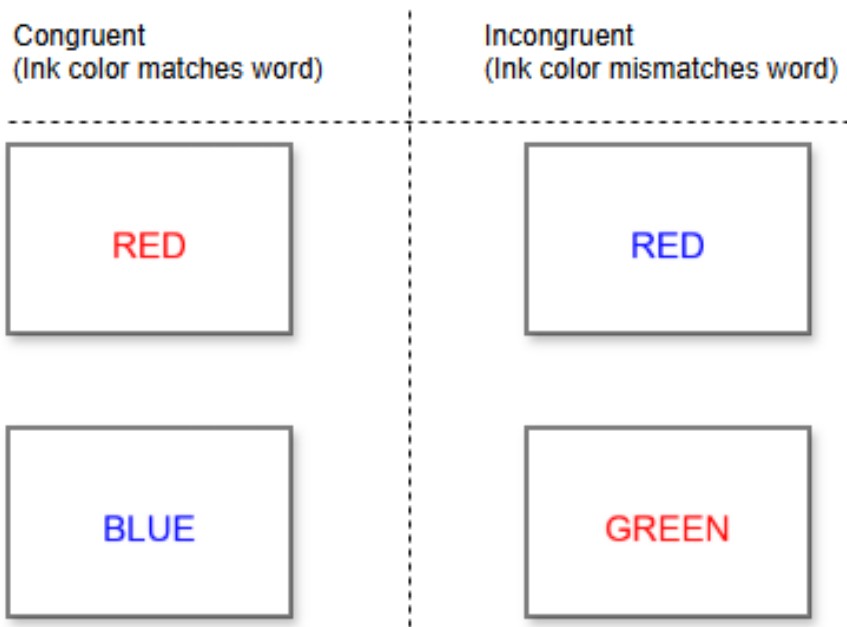

Figure 8: Examples from the dataset showing congruent and incongruent stimuli.

| Prompt family | Congruent (n=10) | Incongruent (n=90) |
|---|---|---|
| Word-oriented (*"The text says X"*) | 10/10 (100%) | 88/90 (**97.8%**) |
| Ink-oriented (*"Written in X color"*) | 10/10 (100%) | 18/90 (**20.0%**) |

Table 9: Family-specific accuracies. Reporting both views avoids conflating the *decision rule* (winner-takes-all) with *family-specific* correctness.

Taken together, Tables 8 and 9 reconcile the two perspectives used in the paper: the overall decision leans to the *word* on most incongruent trials, and—when families are evaluated separately—text prompts are correct far more often than color prompts.

## B.2 VISUAL MANIPULATION EFFECTS

We probe whether changing legibility or perceptual salience shifts CLIP's preference. Four manipulations are considered: font size, font weight, contrast, and pseudowords. Figures 9–13 show exemplars; aggregated numbers are summarized below.

| Manipulation | Setting | Text match | Color match | Notes |
|---|---|---|---|---|
| Font size | 48 pt | 97% | 13% | Readable; text dominates |
| | 72 pt | 95% | 15% | |
| | 90 pt | 89% | 21% | |
| | 108 pt | 77% | 33% | Overflow reduces readability |
| Font weight | Bold | 89% | 21% | Minor effect on color |
| | Normal | 88% | 22% | |
| | Light | 89% | 21% | |
| | Narrow | 97% | 13% | Highest text bias |
| Contrast (incongruent) | High | 87.78% | 12.22% | |
| | Medium | 83.33% | 16.67% | Shift from text→color as contrast drops |
| | Low | 67.78% | 32.22% | |
| | Same | 0.00% | 100.00% | Text unreadable |
| Tone variation (incongruent) | All tones | 86.40% | 3.60% | Lower chroma reduces color salience; text dominates |
| Pseudowords (incongruent) | — | Color match ≈ 78% | | No lexical cue; color used |

Table 10: Effect of visual manipulations (aggregated across colors). Values mirror the panels in Figs. 9–13; tone-variation exemplars are in Fig. 12.

CLIP "reads when it can": high legibility (smaller size, heavier weight, strong contrast) reliably yields text-aligned decisions; suppressing legibility (overflow, low/same contrast, pseudowords) flips decisions toward ink color. The trend is monotonic under contrast (High→Same) and only partially attenuated by making text visually larger/heavier.

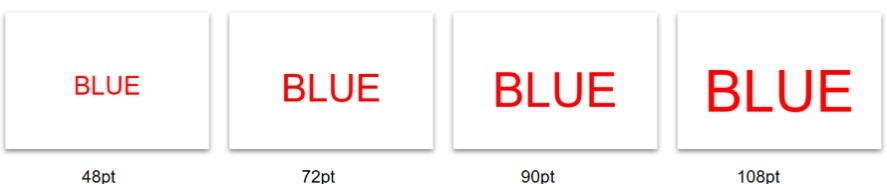

Figure 9: Font size exemplars and aggregate trend (Table 10).

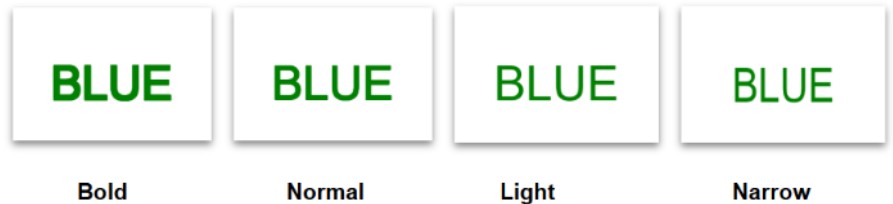

Figure 10: Font weight exemplars and aggregate trend (Table 10).

## B.3 PSEUDOWORD RESPONSE CATEGORIES

Since pseudowords carry no semantic meaning, strictly speaking, there are no true "congruent" cases. We therefore break down responses into four categories:

- **Color Match (ink-based):** The model outputs the correct ink color, independent of the pseudoword form. Example: *GRIV* printed in green ink, model answers "green."

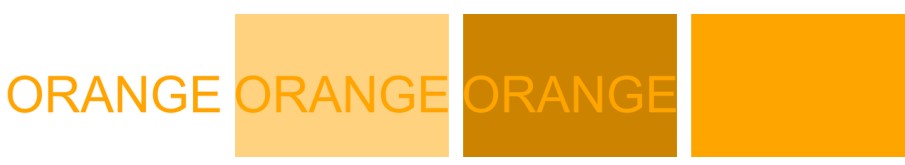

Figure 11: Contrast manipulation exemplars; see Table 10.

BLUE BLUE BLUE BLUE BLUE BLUE BLUE BLUE BLUE BLUE

BLUE BLUE BLUE BLUE BLUE BLUE BLUE BLUE BLUE BLUE

BLUE BLUE BLUE BLUE BLUE BLUE BLUE BLUE BLUE BLUE

BLUE BLUE BLUE BLUE BLUE BLUE BLUE BLUE BLUE BLUE

Figure 12: Tone variations (examples).

GREL GREL GREL GREL GREL GREL

GREL GREL GRIV GRIV GRIV GRIV

GRIV GRIV GRIV GRIV GRIV GRIV

GRON GRON GRON GRON GRON GRON

Figure 13: Pseudoword exemplars; no lexical cue leads to color-based decisions.

- **Form–Color Aligned:** In some cases, a pseudoword resembles a real color word (e.g., *BLIR* ∼ "blue"). When printed in that same color, both cues align. Earlier drafts labeled this as "congruent," but here we adopt the more precise term *Form–Color Aligned*.

- **Text-Biased (form-based):** The model treats the pseudoword as if it were a real color word, even when the ink color differs. Example: *GRIV* printed in red ink, model answers "green."

- **Other Errors:** Predictions that match neither ink color nor pseudoword form. Example: *GRIV* printed in red ink, model answers "purple."

NONSENSE-WORD CONTROL EXPERIMENT ("ZARP")

To clarify whether CLIP's behavior in the Stroop setting reflects genuine color understanding or simply an OCR-driven preference for whatever text appears in the image, we conducted the reviewer's proposed control experiment using the pseudoword *zarp*. We regenerated the entire Stroop set with the word *zarp* rendered in each of the ten ink colors, and—critically—added *zarp* to the list of candidate color labels so that the model could select it as a legitimate answer.

| Category | Count | Accuracy (%) |
|---|---|---|
| Color Match (ink-based) | 389 | 77.8 |
| Form–Color Aligned | 49 | 9.8 |
| Text-Biased (form-based) | 28 | 5.6 |
| Other Errors | 34 | 6.8 |
| **Total** | **500** | 100 |

Table 11: Aggregated pseudoword results across all colors ($N = 500$). No true "congruent" cases exist; **Form–Color Aligned** refers to coincidental alignment of pseudoword form and ink color.

The outcome was unambiguous: across all ink colors, CLIP chose *zarp* in **100%** of trials, resulting in **0%** ink accuracy. In other words, even when the printed token carries no meaning, CLIP treats it as a valid color category purely because it is visually present in the image.

This result makes the underlying mechanism clear. When faced with an unfamiliar string, CLIP does not rely on color semantics at all; instead, it aligns the image embedding with the closest matching text embedding in a strongly OCR-like manner. This complements the main paper's representational analysis: the word-over-color effect does not reflect semantic reasoning about colors, but rather a structural bias in CLIP's embedding geometry that privileges textual features over chromatic ones.

## C  CLIP Latent Space Analyses

To complement the behavioral findings, this appendix examines CLIP's internal image representations. We ask whether the text-over-color preference observed in Sec. 4 is also reflected *in the geometry* of the embedding space. We analyze (i) unguided shape vs. color dominance, (ii) modality-specific contributions using Representational Dissimilarity Matrices (RDMs), (iii) concept sensitivity via clustering, and (iv) low-dimensional structure with UMAP.

### C.1  Word vs. Ink Dominance in Embedding Space

Unlike the prompt-based behavioral readout, here we take a neutral setup: image embeddings are compared only to bare color-word text embeddings (e.g., "red", "blue") without sentence framing. For each Stroop image we compute cosine similarity to two anchors—the written *word* and the *ink color*—and label the case as *word-aligned* or *ink-aligned* accordingly.

Across the 90 incongruent images, CLIP aligns with the *written word* in 87 cases and with the *ink color* in 3 (96.7% vs. 3.3%). Congruent images (10/10) align equally with both concepts as expected.

Even without prompt steering, CLIP's image embeddings sit closer to *textual* concepts than to ink colors when the two conflict.

### C.2  Modality Impact via Representational Dissimilarity Matrices

To complement the behavioral analyses, we used Representational Dissimilarity Matrices (RDMs) to directly probe how CLIP's embeddings reorganize when *word* or *ink* information is added. We build RDMs (pairwise cosine dissimilarity) for three inputs: **Ink-only** (solid backgrounds), **Word-only** (grayscale words), and **Word+Ink** (colored words). Modality-specific contributions are isolated by subtraction:

$$\Delta\text{Color} = \text{RDM(Word+Ink)} - \text{RDM(Shape)}, \quad \Delta\text{Shape} = \text{RDM(Word+Ink)} - \text{RDM(Color)}.$$

Adding *word* reorganizes the space more sharply than adding *ink color*, mirroring the behavioral text/shape preference.

#### Clustering and Dissimilarity Grouping

We quantify per-color sensitivity to added text by the mean change in dissimilarity between Ink-only and Word+Ink embeddings, then cluster colors via agglomerative (Ward) linkage on cosine distance.

Text dominance is *systemic but not uniform*: CLIP's reliance on shape varies with the color concept.

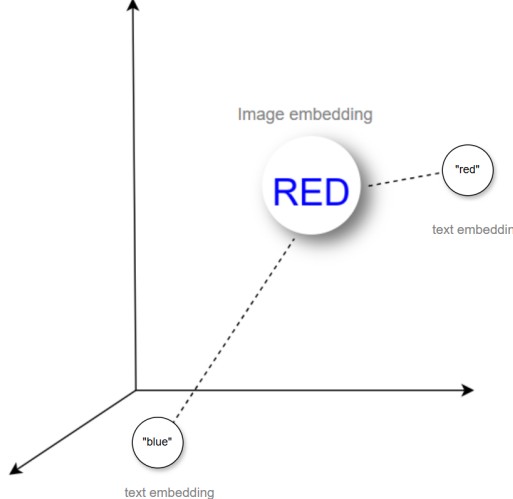

Figure 14: Unguided similarity test: the image embedding (e.g., "RED" written in blue) is compared to the concept anchors *red* and *blue* in text space; the closer anchor indicates the dominant modality.

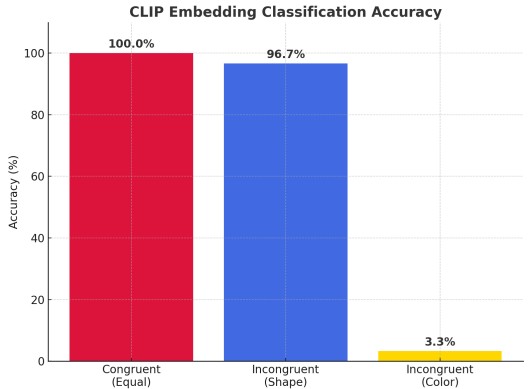

| Condition | Ink | Word |
|---|---|---|
| Incongruent | 3 | 87 |
| Congruent | 10 | 10 |

Figure 15: Alignment counts for congruent vs. incongruent.

Table 12: Modality dominance under unguided comparisons.

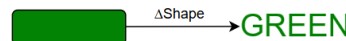

Figure 16: Design for modality-isolating RDM comparisons. ΔWord/Shape asks "what changes when text is added?", ΔInk/Color asks "what changes when color is added?"

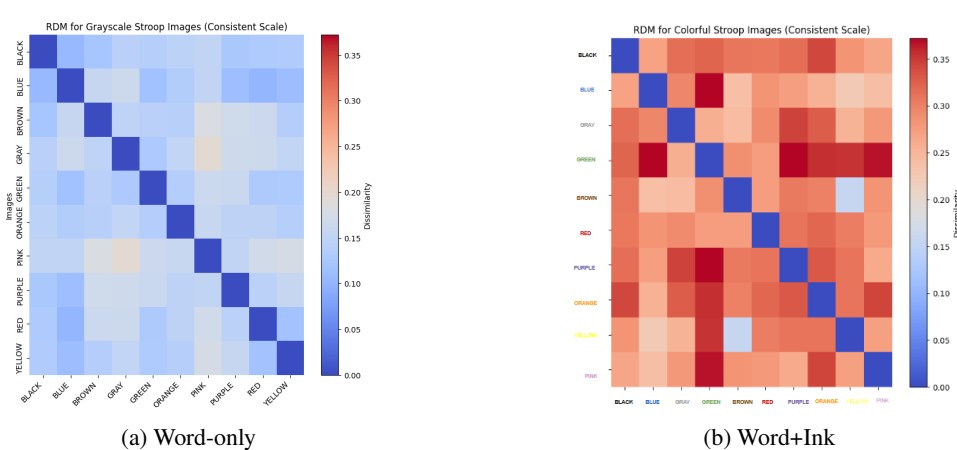

Figure 17: RDMs for grayscale vs. colorful Stroop images.

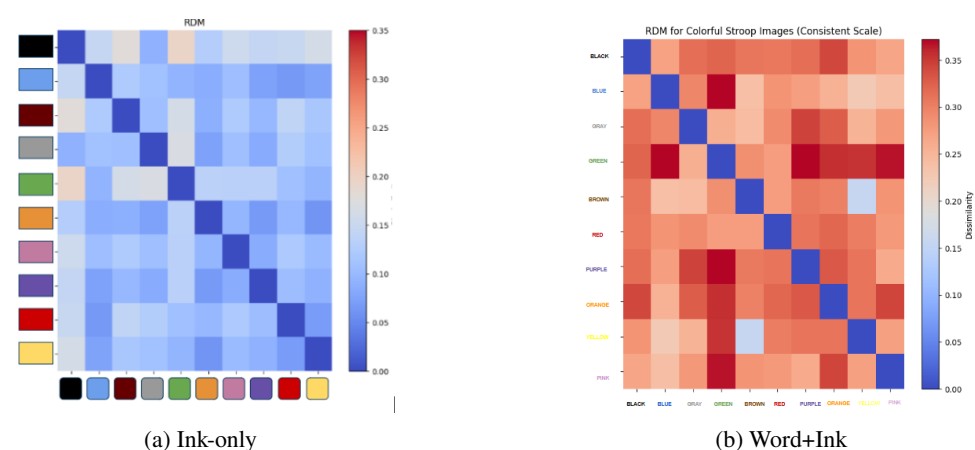

Figure 18: RDMs for solid-color vs. colorful text images.

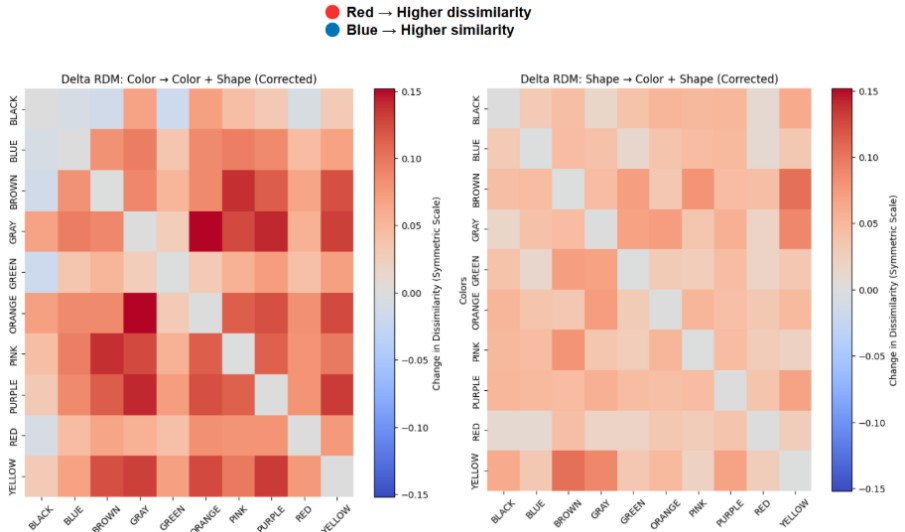

Figure 19: Delta RDMs. Left: ΔWord (add text to color); Right: ΔInk (add color to text). Stronger, more localized structure from text; color induces weaker, diffuse changes.

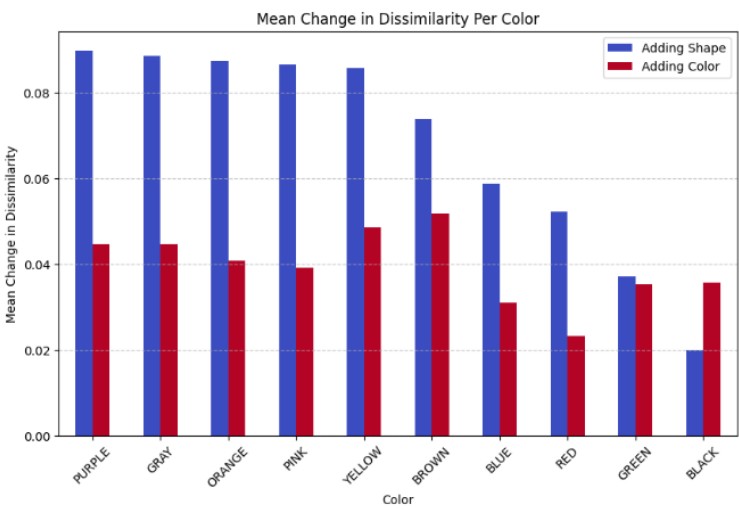

Figure 20: Mean dissimilarity change when adding shape (blue) vs. color (red) per hue. Shape dominates across most colors; magnitude varies by concept.

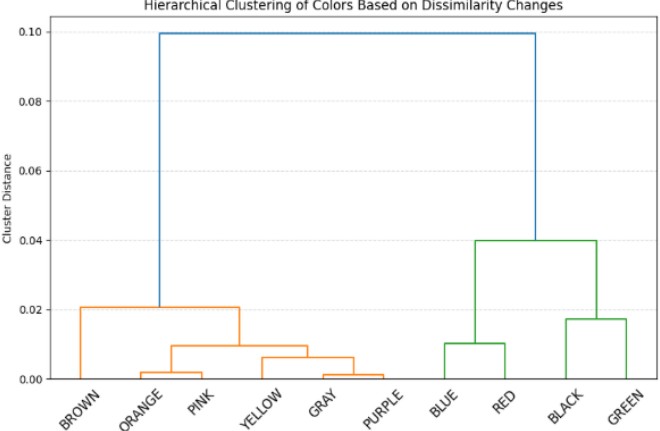

Figure 21: Hierarchical clustering by shape-induced change. Colors differ in how strongly text reshapes their embeddings (e.g., purple/gray/orange high; black/green low).

### VISUALIZING MODALITY SEPARATION WITH UMAP

To better understand how CLIP organizes visual and textual inputs, we use UMAP McInnes et al. (2018) to project embeddings into two dimensions. UMAP preserves local neighborhood structure and reveals clustering patterns, but as a dimensionality-reduction method it cannot fully preserve the geometry of the original space. The plots should therefore be interpreted as an illustrative view rather than an exact map of the latent space.

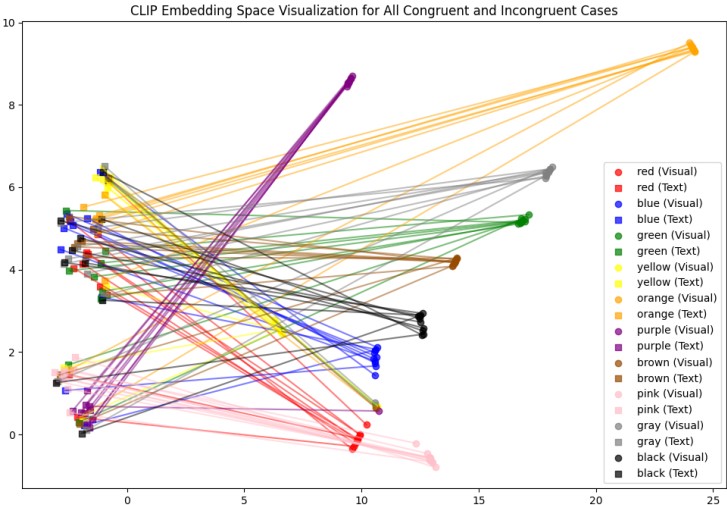

Figure 22: UMAP view. Text prompts cluster tightly by semantics; image embeddings group primarily by the *written word* rather than by ink color.

Both modalities organize around *meaning*, not hue—again consistent with behavioral results.

### C.5  REPRESENTATIVE CLIP OUTPUTS

Exemplar cosine-similarity outcomes under incongruent stimuli (winner among family-specific scores). Despite being asked for ink (Sec. 4), CLIP tends to output the textual label.

| Stimulus | Ground Truth (Ink) | Top-1 Text | Top-1 Color | Bias |
|---|---|---|---|---|
| "RED" in blue ink | Blue | Red (0.89) | Blue (0.12) | Textual |
| "GREEN" in pink ink | Pink | Green (0.76) | Pink (0.24) | Mixed |
| "BLACK" in yellow ink | Yellow | Black (0.82) | Yellow (0.10) | Textual |

Table 13: Example CLIP similarity outputs on incongruent Stroop stimuli (cosine scores in parentheses).

### KEY TAKEAWAYS

- **Unguided dominance:** In incongruent cases, 96.7% of image embeddings align with the *written word* (Fig. 15, Tab. 12).

- **RDM evidence:** $\Delta$Word induces stronger, localized structural changes than $\Delta$Ink (Fig. 19).

- **Concept sensitivity:** Text dominance varies by hue (clustering in Fig. 21).

- **Low-D view:** UMAP shows tight semantic clusters for text and word-driven grouping for images (Fig. 22).

# D LATENT INTERVENTION ANALYSIS: CHUNK EXTRACTION, NORMS, AND ROBUSTNESS

This appendix provides extended results and implementation details for the latent intervention experiments: subpopulation-based chunk extraction, cosine-shift success, vector norms, and robustness to prompt variation.

## D.0 EXPERIMENTAL SETUP AND EVALUATION PROTOCOL

**Embeddings.** We use CLIP's image encoder to obtain $d$–dimensional, $\ell_2$–normalized image embeddings. After any edit, we re-project to the unit sphere: $\mathbf{E}' \leftarrow \mathbf{E}'/\|\mathbf{E}'\|_2$ so that cosine similarity remains well-defined.

**Concept sets.** Chunks are built from *congruent* images only (10 per class; App. A). For each source–target pair we intervene on 90 *incongruent* sources $\times$ 9 targets.

Cosine similarity is computed to two prompt families (App. A): color–oriented and text–oriented. The shift metric is
$$\Delta = \mathrm{sim}(\mathbf{E}', \text{target}) - \mathrm{sim}(\mathbf{E}', \text{source}),$$
and we count a *success* if $\Delta > 0$.

Unless noted, edits use scale $\alpha = 1$. For completeness we also report robustness to $\alpha$ (see Sec. D.6).

## D.1 SUBPOPULATION-BASED CHUNK EXTRACTION

To manipulate semantic concepts we derive directional vectors (*chunks*) representing interpretable transitions (e.g., "red" $\rightarrow$ "blue"). For each concept $c$, we compute per-dimension variance across its $N$ embeddings; cluster the variance values with $k$–means ($k = 2$); and keep the *low-variance* cluster as a binary mask $M_c \in \{0, 1\}^d$. Stable means are
$$\mu_c^{\text{stab}} = \frac{1}{N} \sum_i (M_c \odot \mathbf{E}_i^{(c)}),$$
and the chunk vector is
$$\bar{c}_{s \to t}^{\text{stab}} = \mu_t^{\text{stab}} - \mu_s^{\text{stab}}.$$

We apply edits with optional scale $\alpha$:
$$\mathbf{E}' = \mathrm{norm}\left(\mathbf{E} - \alpha\,\mu_s^{\text{stab}} + \alpha\,\mu_t^{\text{stab}}\right), \quad \text{where } \mathrm{norm}(\cdot) \text{ reprojects to the unit sphere.}$$

## D.2 SUBPOPULATION-BASED INTERVENTION RESULTS

Table 14: Success and shift for subpopulation-based interventions

| Intervention | Success rate | Mean $\Delta$ |
|---|---|---|
| Color | 36.7% | $-0.0017$ |
| Text | 100.0% | $+0.0934$ |
| Combined | 100.0% | $+0.1172$ |

## D.3 CHUNK VECTOR NORM STRENGTH

## D.4 PROMPT ROBUSTNESS

## D.5 DECOMPOSING COMBINED INTERVENTIONS

## D.6 ROBUSTNESS TO EDIT SCALE $\alpha$

We swept $\alpha \in \{0.25, 0.5, 1.0, 1.5, 2.0\}$. Text and combined edits improve monotonically up to $\alpha \approx 1.5$ and then saturate; color-only edits remain near chance and become unstable for $\alpha > 1.5$ (occasional regressions).

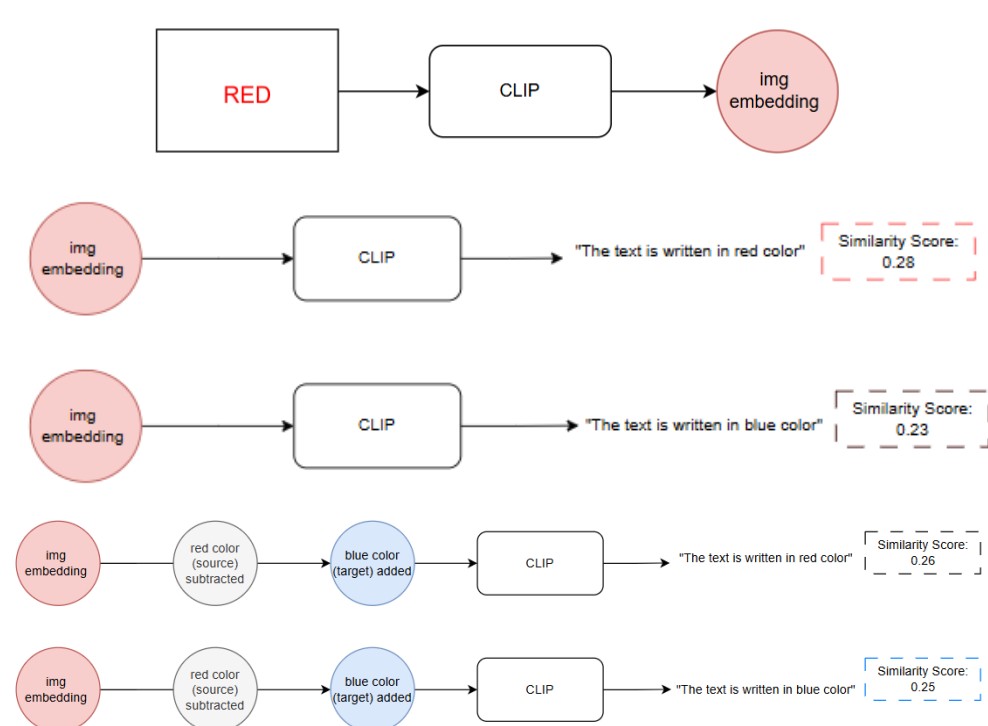

Figure 23: Subpopulation-based intervention pipeline (RED→BLUE). After editing we always re-normalize embeddings.

Table 15: Chunk $\ell_2$ norms (color vs. text); mean across pairs shows ∼2.1× stronger text directions

| Source→Target | Color norm | Text norm | Notes |
|---|---|---|---|
| red→blue | 2.53 | 5.38 | |
| red→pink | 3.60 | 6.38 | |
| orange→brown | 2.09 | 6.61 | (min color) |
| pink→green | 3.86 | 6.49 | (max color) |
| **Mean** | **2.99** | **6.36** | |

Table 16: Effect of prompt variants on color-only intervention accuracy

| Variant | Template | Accuracy |
|---|---|---|
| color | The text is written in {color} color | 36.7% |
| just | {color} | 41.1% |
| pure | The text is purely {color} | 41.1% |
| long | The color of the text is purely {color} | 38.9% |

Table 17: Contribution of color chunk within combined edits

| Metric | Value |
|---|---|
| Mean $\Delta$ (Combined vs. Text-only) | +0.0069 |
| Cases with $|\Delta| < 0.01$ | 60% |
| Mean color-chunk norm | 3.04 |

## D.7 SUMMARY

- CLIP's latent space is **modular for text**, but **entangled for color**. Text chunks are $\sim 2.1\times$ stronger in norm and yield 100% success; color-only remains weak even after subpopulation filtering.

- Subpopulation filtering yields stable chunk vectors, but color interventions still succeed in only 36.7% of cases with near-zero mean shifts.

- Prompt variants provide small gains; scaling $\alpha$ mainly benefits text/combined edits.

- Combined success is driven almost entirely by the text component (Sec. D.5).

# E  LABEL MAPPING AND POST-PROCESSING RULES

For evaluation, raw model outputs or similarity scores are mapped into four categories: **Word Match**, **Ink Match**, **Both**, and **Neither**. The mapping procedure ensures consistency across CLIP/SigLIP (similarity-based) and generative VLMs (free-form text outputs).

- **Word-oriented prompts** ("The text says X"): If the top-1 prediction is X, we assign a *Word Match*.

- **Ink-oriented prompts** ("The text is written in X color"): If the top-1 prediction is X, we assign an *Ink Match*.

- **Tie-handling (similarity-based models)**: When cosine similarities are equal, ties are resolved by a fixed priority: Word Match > Ink Match > Neither.

- **Generative VLM outputs**: Free-form text is normalized (lowercased, trimmed, and mapped to the set of 10 basic colors {red, blue, green, yellow, orange, pink, purple, black, brown, gray}). Non-matches are labeled as *Neither*.

- **Qwen2-VL prompts:** responses are already constrained to a single lowercase color word; we still normalize to the 10-color lexicon and map out-of-vocabulary items to *Neither*.

## E.1  GENERATIVE VLM SCORING PROCEDURE (TSR)

For completeness and to ensure comparability across all evaluated models, we report here the procedure used to score free-form outputs from generative VLMs. All responses are processed through a deterministic, rule-based normalization pipeline: outputs are lowercased, stripped of punctuation, and matched against a fixed lexicon of ten canonical color labels {red, blue, green, yellow, orange, pink, purple, black, brown, gray}. Responses that do not match any valid label are assigned the category *Neither*. No LLM-as-a-Judge or heuristic prompt interpretation is used.

Because instruction-tuned generative models are expected to follow formatting constraints, we additionally compute a **Template Success Rate (TSR)**, defined as the fraction of outputs that adhere to the required format of a single lowercase color word. This allows us to separate genuine Stroop behavior from potential instruction-following failures.

- **Qwen2-VL-7B-Instruct**: TSR = 100%. All responses were correctly formatted single-token color labels. Nevertheless, decisions were strongly dominated by the written word (97% word-match vs. 13% ink-match), indicating that the observed Stroop pattern is not attributable to formatting or parsing issues.

- **LLaVA-1.5**: TSR = 92%. Among parseable outputs, the model exhibited a similar bias (88% word-match vs. 12% ink-match), again showing that instruction compliance does not eliminate word dominance.

For caption-driven or non-instruction-tuned models (e.g., BLIP-2, Kosmos-2), strict templating is not meaningful: these models are not designed to output single-token answers. Applying TSR would therefore conflate instruction capability with Stroop sensitivity. We instead evaluate such models using their unconstrained outputs passed through the same normalization and lexicon mapping pipeline described above.

Taken together, these results confirm that the Stroop-style word preference in generative VLMs persists even when instructions are fully obeyed and outputs are perfectly formatted. The bias therefore reflects a genuine property of their underlying semantic representations rather than an artifact of output formatting.

# F  SigLIP-2 Behavioral Results

This appendix reports full results for SigLIP-2 under the Stroop-style probes described in Section 4. While SigLIP-2 introduces architectural improvements over CLIP, its behavior under multimodal conflict remains strongly word-dominant. Below we provide condition-wise breakdowns.

## F.1  Congruent vs. Incongruent Cases

| Condition | Text Accuracy (%) | Color Accuracy (%) |
|---|---|---|
| Congruent | 100.0 | 100.0 |
| Incongruent | 100.0 | 5.6 |

Table 18: SigLIP-2 behavioral accuracy on congruent and incongruent Stroop stimuli.

## F.2  Font Size Variation (48–108 pt)

| Font Size | Text Accuracy (%) | Color Accuracy (%) |
|---|---|---|
| 48 pt | 100.0 | 15.0 |
| 72 pt | 100.0 | 16.0 |
| 90 pt | 100.0 | 16.0 |
| 108 pt | 100.0 | 16.0 |

Table 19: Accuracy by font size for SigLIP-2. Unlike CLIP, larger sizes did not weaken text dominance.

## F.3  Font Weight Variation

| Style | Text Accuracy (%) | Color Accuracy (%) |
|---|---|---|
| Light | 100.0 | 6.7 |
| Normal | 100.0 | 6.7 |
| Bold | 100.0 | 7.8 |
| Narrow | 100.0 | 6.7 |

Table 20: Accuracy by font weight. SigLIP-2 remained insensitive to style; text was always dominant.

## F.4  Contrast Manipulation

| Contrast | Text Win (%) | Color Win (%) | Tie (%) |
|---|---|---|---|
| High | 55.6 | 11.1 | 33.3 |
| Medium | 44.4 | 11.1 | 44.4 |
| Low | 22.2 | 66.7 | 11.1 |
| Same | 33.3 | 11.1 | 55.6 |

Table 21: Preference breakdown for contrast manipulations. SigLIP-2 showed indecision (ties) at medium/same levels, unlike CLIP which switched fully to color when text was unreadable.

## F.5  Pseudowords

Across all manipulations, SigLIP-2 exhibited a rigid Stroop-like bias toward text, with 100% text accuracy under incongruence. Visual manipulations (size, weight, contrast) did not weaken the

| Outcome | Count (%) |
|---|---|
| Ink-oriented prompt wins | 335 (67%) |
| Word-oriented prompt wins | 139 (28%) |
| Tie | 26 (5%) |

Table 22: SigLIP-2 on pseudoword stimuli. Unlike CLIP, which often clung to text shapes, SigLIP-2 switched reliably to color.

bias, though pseudowords revealed a partial recovery of color accuracy. These results indicate that SigLIP-2, despite training improvements, inherits the same text-over-color preference as CLIP.

## G  MODEL DETAILS

Table 23 summarizes the specific versions of all models used in our experiments, including their backbone, approximate parameter count, training objective, and release source. These choices reflect widely adopted public checkpoints to ensure reproducibility.

| Model | Backbone | Params | Training Objective | Source / Release |
|---|---|---|---|---|
| CLIP | ViT-B/32 + Transformer text encoder | ∼150M | Contrastive (softmax) | Radford et al. (2021) |
| SigLIP-2 | ViT-B/16 (base) + text encoder | ∼150M | Contrastive (sigmoid) | Google, 2024 release |
| BLIP-2 | ViT-g + FlanT5-XL | ∼13B | Q-former + pretrain + instruction tuning | Li et al. (2022) |
| InstructBLIP | ViT-g + Vicuna-7B | ∼7B | Instruction-tuned generation | Dai et al. (2023) |
| Kosmos-2 | Multi-stream transformer (1B) | ∼1B | Multimodal pretrain (text+image) | Microsoft, 2023 |
| LLaVA (v1.6) | CLIP ViT-L/14 + Vicuna-7B | ∼7B | Instruction-tuned generation | Liu et al. (2023) |
| GIT | ViT-B/16 encoder + GPT2-style decoder | ∼400M | Pretrain (captioning) | Wang et al. (2022) |
| Qwen2-VL-7B-Instruct | ViT-based vision encoder + 7B LLM | ∼7B | Multimodal instruction tuning | Wang et al. (2024) |

Table 23: Model configurations used in this study. Parameter counts are approximate; all models are publicly available through their original repositories.

## H  QWEN2-VL-7B-INSTRUCT: EXTENDED SETUP AND RESULTS

We also tested Qwen2-VL-7B-Instruct, a recent 7B-parameter vision–language model released by Alibaba's Qwen team (Wang et al., 2024). Experiments were run with HuggingFace Transformers (v4.56.1) on LRZ HPC using NVIDIA H100 GPUs (80 GB). The model follows a multimodal encoder–decoder design, pairing a ViT-based vision encoder with an instruction-tuned large language model.

To probe Stroop-style conflicts, we used the same synthetic dataset as in App. A (100 stimuli: 10 congruent, 90 incongruent). Two prompt variants were tested:

- **Main:** "You will see a single English word rendered in a colored ink. Ignore the written word and answer ONLY the ink color as one lowercase color name."

- **Alternative:** "What is the ink color of the text in the image? Answer with one lowercase color word only."

Performance on congruent items was perfect under both prompts (100% ink and word alignment). In incongruent cases, however, outcomes differed sharply: the longer instruction yielded only 31.1% ink accuracy, while the shorter variant nearly doubled it to 60.0%. This indicates that even advanced

instruction-tuned VLMs inherit Stroop-style text dominance, but their behavior is sensitive to how the task is phrased.

| Condition | Main prompt (%) | Alt prompt (%) |
|---|---|---|
| Congruent (n=10) | 100.0 | 100.0 |
| Incongruent (n=90) | 31.1 | 60.0 |

Table 24: Qwen2-VL-7B-Instruct Stroop results under two prompts. Prompt choice strongly influences incongruent ink accuracy.

Overall, Qwen2-VL shows somewhat higher sensitivity to ink color compared to CLIP and SigLIP-2, but still defaults to the written word when word and ink conflict. The gap between the two prompt conditions highlights the role of instruction design in shaping model responses.

PROMPT VARIATION ANALYSIS

Given the reviewer's question regarding whether Qwen2-VL's Stroop behavior reflects a representational limitation or an instruction-following issue, we conducted an extended prompt-variation experiment. Building on the two prompts reported in the main appendix (Main vs. Alt), we constructed a broader set of ten minimally differing instructions:

**p1** "What is the ink color of the text in the image? Answer with one lowercase color word only."

**p2** "Identify only the color of the ink used in the image. Respond with one lowercase color word."

**p3** "Given the image, state the ink color of the printed word. Use a single lowercase color name."

**p4** "Look at the picture and name the color of the ink, not the written word. Reply with a lowercase color."

**p5** "Return only the ink color shown in the image. Do not describe anything else."

**p6** "State the ink color of the letters. One lowercase color word only."

**p7** "Ink color only — answer with one simple lowercase color word."

**p8** "What is the color of the letters in this picture? Reply with one lowercase color name."

**p9** "Read the image and give only the ink color as a lowercase color word."

Each prompt was evaluated on the 10 congruent and 90 incongruent Stroop items. Per-prompt ink accuracies are reported below:

| Prompt | Ink Preference (%) |
|---|---|
| p1 | 58.0 |
| p2 | 88.0 |
| p3 | 48.0 |
| p4 | 45.0 |
| p5 | 83.0 |
| p6 | 37.0 |
| p7 | 24.0 |
| p8 | 97.0 |
| p9 | 31.0 |

Table 25: Ink accuracy of Qwen2-VL-7B-Instruct under nine alternative prompt formulations (p1–p9).

The results reveal substantial variability: ink accuracy ranges from as low as 24% (up to 37–48% for several neutral phrasings) to as high as 88–97% for the clearest and most narrowly targeted instructions (prompts p2, p5, p8). This confirms that Qwen2-VL *is capable* of attending to ink color, but only when the prompt is formulated with sufficient specificity and minimal linguistic ambiguity.

These findings address the reviewer's concern directly. The variability does not contradict our core claim: unlike CLIP, whose word-dominance is fixed by the geometry of its embedding space, Qwen2-VL exhibits an *instruction-sensitive* form of text bias. The underlying representation still favors the written word by default, but careful prompting can partially redirect the model's attention toward ink color. Thus the effect is not purely representational nor purely semantic—rather, it reflects how Qwen2-VL balances visual features with instruction-following behavior. This helps explain why different prompt phrasings yield dramatically different outcomes, even though the underlying visual representations remain largely unchanged.

# I    Layer-wise Steering Details for Qwen2-VL and LLaVA

This section provides the full layer-wise steering results for Qwen2-VL-7B and LLaVA-1.6-7B. For each layer, we report three metrics corresponding to (1) color steering, (2) word steering, and (3) combined steering. Steering effectiveness is quantified using the Directional Similarity Shift (DSS):

$$\text{DSS} = \cos(E', \text{target}) - \cos(E, \text{target}),$$

where $E$ and $E'$ denote the original and edited embeddings respectively. An intervention is counted as *successful* if $\text{DSS} > 0$. Success rates below summarize the proportion of samples in which a steering edit moves the embedding toward the intended concept.

## I.1    Qwen2-VL-7B: Full Layer-wise DSS Success Rates

Qwen2-VL shows nearly perfect color steering across all layers, word steering that is initially weak but becomes strong in later layers, and combined steering that succeeds in all layers. After layer 4, both word and color directions stabilize sharply, with layer-to-layer cosine similarity in the range 0.93–0.99.

## I.2    LLaVA-1.6-7B: Full Layer-wise DSS Success Rates

LLaVA preserves visual color information well across layers, producing strong color steering. Word steering remains weak throughout, consistent with the fact that LLaVA injects linguistic features late via a visual projector. Combined steering largely follows the color pattern.

## I.3    Interpretation

Taken together, the layer-wise analyses reveal clear architectural differences across the three models. Qwen2-VL rapidly collapses its visual features into a stable, LLM-like semantic space, which makes both word and color directions highly consistent and easy to steer after only a few layers. LLaVA, in contrast, preserves a more substantial visual pathway: color information remains robust and steerable throughout the network, whereas word-related directions remain comparatively weak because linguistic information is injected later through the projection module. CLIP, as discussed in the main paper, is strongly shaped by its language-aligned contrastive training, producing highly steerable word directions but fragile and entangled color directions. These architectural signatures help explain the behavioral differences observed across models and clarify why word and color cues contribute differently to each system's internal representations.

## I.4    Large-Scale Stroop Dataset (23k Images)

To complement the controlled 100-image Stroop set, we generated a large-scale variant consisting of 23,338 images. While congruent cases were kept clean and minimal (10 total), the remaining 23,328 incongruent stimuli were produced using systematic variations in background texture, tone, saturation, brightness, and mild spatial perturbations. These manipulations ensure that models cannot rely on template memorization and instead must perform genuine color recognition under appearance diversity.

Each stimulus was rendered using the same 10 ink-color categories as the main dataset. For evaluation, CLIP was tested with the standard ink- and word-oriented prompt sets. The model maintained

Table 26: Qwen2-VL-7B: Layer-wise DSS Success Rates

| Layer | Color | Word | Combined |
|-------|-------|-------|----------|
| 0 | 0.889 | 0.700 | 0.856 |
| 1 | 1.000 | 0.844 | 1.000 |
| 2 | 1.000 | 0.367 | 1.000 |
| 3 | 1.000 | 0.500 | 1.000 |
| 4 | 1.000 | 0.644 | 1.000 |
| 5 | 1.000 | 0.522 | 1.000 |
| 6 | 1.000 | 0.656 | 1.000 |
| 7 | 1.000 | 0.833 | 1.000 |
| 8 | 1.000 | 0.756 | 1.000 |
| 9 | 1.000 | 0.856 | 1.000 |
| 10 | 1.000 | 0.767 | 1.000 |
| 11 | 1.000 | 0.833 | 1.000 |
| 12 | 1.000 | 0.756 | 1.000 |
| 13 | 1.000 | 0.778 | 1.000 |
| 14 | 1.000 | 0.933 | 1.000 |
| 15 | 1.000 | 0.922 | 1.000 |
| 16 | 1.000 | 0.900 | 1.000 |
| 17 | 1.000 | 0.933 | 1.000 |
| 18 | 1.000 | 0.956 | 1.000 |
| 19 | 1.000 | 0.889 | 1.000 |
| 20 | 1.000 | 0.833 | 1.000 |
| 21 | 1.000 | 0.767 | 1.000 |
| 22 | 1.000 | 0.789 | 1.000 |
| 23 | 1.000 | 0.889 | 1.000 |
| 24 | 1.000 | 0.800 | 1.000 |
| 25 | 1.000 | 0.944 | 1.000 |
| 26 | 1.000 | 0.911 | 1.000 |
| 27 | 1.000 | 0.889 | 1.000 |
| 28 | 1.000 | 0.956 | 1.000 |
| 29 | 1.000 | 0.967 | 1.000 |
| 30 | 1.000 | 0.978 | 1.000 |
| 31 | 1.000 | 0.978 | 1.000 |

Table 27: LLaVA-1.6-7B: Layer-wise DSS Success Rates

| Layer | Color | Word | Combined |
|-------|-------|------|----------|
| 0 | 0.678 | 0.411 | 0.678 |
| 1 | 0.533 | 0.333 | 0.522 |
| 2 | 0.767 | 0.344 | 0.778 |
| 3 | 0.811 | 0.322 | 0.789 |
| 4 | 0.889 | 0.378 | 0.889 |
| 5 | 0.822 | 0.300 | 0.800 |
| 6 | 0.756 | 0.422 | 0.722 |
| 7 | 0.800 | 0.333 | 0.744 |
| 8 | 0.733 | 0.311 | 0.711 |
| 9 | 0.689 | 0.400 | 0.722 |
| 10 | 0.722 | 0.356 | 0.678 |
| 11 | 0.789 | 0.378 | 0.767 |
| 12 | 0.856 | 0.444 | 0.811 |
| 13 | 0.844 | 0.322 | 0.878 |
| 14 | 0.800 | 0.300 | 0.833 |
| 15 | 0.800 | 0.289 | 0.778 |
| 16 | 0.822 | 0.444 | 0.844 |
| 17 | 0.833 | 0.300 | 0.844 |
| 18 | 0.811 | 0.278 | 0.822 |
| 19 | 0.756 | 0.467 | 0.800 |
| 20 | 0.756 | 0.456 | 0.811 |
| 21 | 0.689 | 0.433 | 0.722 |
| 22 | 0.744 | 0.789 | 0.811 |
| 23 | 0.944 | 0.733 | 0.967 |

extremely high word accuracy (99.5%) while ink accuracy remained low (15.9%), reproducing the strong Stroop-style word bias observed in the main paper. Full sampling parameters, rendering functions, and the generation script are included in the project repository and will be released upon acceptance.

## J  FLUX-GENERATED REALISTIC MULTIMODAL CONFLICT SET

This appendix reports the full set of multimodal conflict images generated using the FLUX.1 model, together with CLIP's similarity scores under word- and color-oriented prompts. Unlike the controlled Stroop stimuli used in the main paper, these examples capture realistic forms of semantic contradiction that arise in everyday visual interfaces (e.g., signage, UI icons, safety symbols, battery indicators, or traffic signals). Each image was paired with two prompts—the written word (semantic cue) and the dominant color (visual cue)—allowing us to directly test whether CLIP prioritizes textual or visual information when the two conflict.

Overall, CLIP exhibits a mixed pattern: text-heavy or OCR-like stimuli tend to trigger strong word dominance, whereas chromatically salient or icon-based stimuli shift the model toward color-based decisions. Table 28 summarizes ten representative cases from our evaluation set, showing both the raw similarity scores and the resulting decision for each conflict instance. These examples illustrate that the Stroop-style word bias documented in the synthetic setting generalizes only partially: in realistic conflict scenarios, visual salience and icon semantics can override the written word.

## K  EXTENDED RELATED WORK

A parallel line of work discusses bias in multimodal AI, noting that training distributions can encourage overreliance on written words over visual appearance (Yuksekgonul et al., 2023). Pezeshkpour et al. (Pezeshkpour et al., 2025) similarly find that VLMs generally lean more heavily on textual cues when visual and textual information conflict, and Vo et al. (Vo et al., 2025) highlight that text-based

Table 28: **FLUX-generated multimodal conflict examples.** CLIP's decision reflects either the textual or visual cue depending on saliency.

| Image | Visual Description | word_sim | color_sim | Decision |
|---|---|---|---|---|
|  | STOP (green background) | 0.17 | 0.83 | COLOR |
|  | Low Battery (green icon) | 0.03 | 0.96 | COLOR |
|  | DO NOT ENTER (green sign) | 0.9469 | 0.0530 | WORD |
|  | DO NOT ENTER (red sign) | 0.9953 | 0.0047 | WORD |
|  | EXIT (red sign) | 0.9988 | 0.0011 | WORD |
|  | OFF (green neon) | 0.0020 | 0.9979 | COLOR |
|  | 120 speed limit (blue) | 0.9992 | 0.0007 | WORD |
|  | GO (red traffic light) | 0.2223 | 0.7777 | COLOR |
|  | CONNECTED (red crossed-out wifi) | 0.00015 | 0.9998 | COLOR |
|  | LEFT (arrow pointing right) | 0.9445 | 0.0554 | WORD |

bias can appear even in visual tasks whenever text cues are present. Prior findings largely establish *that* word dominance occurs; our study complements them by examining *why* it occurs. We couple a controlled Stroop-style evaluation with latent-space analysis, linking output-level failures to the structure and steerability of internal representations. This connection clarifies when the written word overrides the ink color and points to concrete levers for improving visual grounding.

The classic Stroop Effect (Stroop, 1935) provides the psychological basis: when word and ink color conflict, people tend to read the *word* rather than name the *ink color*. Recent work has ported this idea to VLMs (Arias et al., 2024), documenting a comparable **word bias** when images contain readable strings. Most of these analyses, however, are deliberately *behavioral*: the model is treated as a black box, conflict is induced at the stimulus level, and conclusions are drawn from outputs alone. What remains underexplored is the representational basis of such dominance—whether and how the learned representation space makes one modality easier to privilege than the other. In contrast, our study goes beyond behavioral outputs: (i) we evaluate a broad family of both contrastive and generative VLMs rather than focusing on CLIP alone; (ii) we systematically manipulate legibility to test the robustness of word dominance; and (iii) we directly analyze the representation space (via RDMs and UMAP) and perform subpopulation-based latent interventions. This dual behavioral–representational approach allows us to explain not only *that* word dominance occurs, but also *why* it arises, revealing structural asymmetries between word and ink-color directions in VLMs.

