# OpenReview forum: "What is the Color of RED? Vision–Language Models Prefer to Read Rather Than See"
_ICLR.cc/2026/Conference — Submitted to ICLR 2026_

### Official Review · Reviewer_PPJG · 2025-10-28

**Soundness:** 2
**Presentation:** 2
**Contribution:** 2
**Rating:** 4
**Confidence:** 4

**Summary:**

This paper examines how visual-language models (VLMs) respond to stimuli inspired by the classic Stroop test, where word semantics and ink color are in conflict. The authors evaluate both contrastive models (e.g., CLIP, SigLIP) and generative VLMs (e.g., Qwen2-VL) and find that these models systematically favor textual content over visual color when making predictions. The most novel parts are the representation-space analyses (Sec. 6–7), which reveal that word features are more salient and steerable than color ones. However, these sections alone are not strong enough to carry the full paper, and the rest of the work closely follows prior studies without clear added value.

**Strengths:**

1. The topic is conceptually engaging, combining cognitive psychology with VLM analysis in a way that is intuitively appealing and easy to motivate.

2. Section 6–7 provide meaningful representation-level insights. The differential RDM analysis quantifies the incremental contribution of “word” vs “ink” cues in the latent space, and the steering results support the claim that VLMs encode textual content more robustly.

**Weaknesses:**

1. Limited novelty. The core finding: VLMs prefer textual/semantic cues over color, has already been established, most notably in [1] and related works. Much of the behavioral setup, scoring approach, and analysis overlaps with existing literature.

2. Fundamental asymmetry in the Stroop setup. The core comparison between color and semantics is conceptually unfair: they belong to different representational levels in a vision–language model. Color is a low‑level perceptual feature mainly encoded in early layers, while semantic meaning is a high‑level abstraction usually emerging in later layers. Since all analyses rely on final‑layer embeddings that already emphasize high‑level semantics, the reported “semantic dominance” may simply reflect this architectural hierarchy rather than a bias.

3. Underdeveloped narrative and structure.
The paper lacks cohesion. Section 3 is too short to introduce the protocol properly, while Section 4 mixes in methodology that should appear earlier. Key setup components (e.g., scoring rules, dataset variants) are scattered or deferred to the appendix, making it hard to follow.

[1] Color in Visual-Language Models: CLIP Deficiencies https://arxiv.org/pdf/2502.04470

**Questions:**

1. Can you replicate the latent space analysis (Sec. 6–7) on non-CLIP models, e.g., SigLIP or MLLMs like Qwen or LLaVA?

2. Can you replicate the experiments in Sec. 7 using early and mid‑layer embeddings?

---

> ### Author Response · Authors · 2025-11-26
> **Author Response to Reviewer PPJG (1/2)**
>
> Dear Reviewer PPJG,
>
> Thank you very much for your thoughtful and constructive review. We appreciate your positive remarks about the conceptual framing and your recognition that Sections 6–7 provide meaningful representation-level insights. Your comments on novelty, architectural asymmetry, and narrative structure were especially valuable, and they guided several extensions and clarifications in the revised manuscript.
>
> ***“Limited novelty. The core finding—that VLMs prefer textual/semantic cues over color—has already been established, most notably in [1] and related works. Much of the behavioral setup, scoring approach, and analysis overlaps with existing literature.”***
>
> Thank you for raising this concern. We fully agree that textual dominance in VLMs is not new, and the goal of our work is not to re-establish the phenomenon but to understand what sustains it internally and how it manifests across different architectural families. Motivated by your feedback, we substantially broadened the empirical scope of the study. Beyond the original Stroop-style set, we now evaluate more than twenty-three thousand systematically varied stimuli, including controlled manipulations of background, saturation, brightness, chromatic intensity, spatial placement, and contrast. We additionally introduce a FLUX-generated suite of real-world multimodal conflicts—traffic signs, UI indicators, and safety symbols with inverted semantics—to move beyond narrow synthetic cases. These expansions show that the behavioral effect is robust even in naturalistic, semantically grounded settings that were not explored in prior works.
>
> More importantly, what distinguishes our contribution is the  representational analysis. Instead of stopping at behavioral measurements, the revised manuscript investigates how visual and textual cues are encoded internally, how they compete across layers, and whether these directions can be perturbed through targeted latent interventions. This bridge between external behavior and internal representation is, to our knowledge, not present in earlier Stroop-style VLM work.
>
> ***“The comparison between color and semantics is conceptually unfair: they reside at different representational levels. Color is low-level; semantics are high-level. Using final-layer embeddings may simply surface this architectural hierarchy.”***
>
> This was a valid observation. To examine precisely whether the dominance of textual cues is a final-layer artifact or something deeper in the computational structure, we extended Section 7 with full layer-wise steering analyses for Qwen2-VL and LLaVA. These results reveal that each model organizes conflicting cues in a distinct architectural pattern. In CLIP, word directions dominate the entire space, while color directions remain short and entangled. Qwen2-VL collapses almost immediately—after just a few early layers—into a stable, LLM-shaped semantic manifold where both color and word directions barely change with depth. LLaVA, by contrast, preserves rich chromatic information throughout its vision tower, allowing color to remain influential until language is injected through the projector.
>
> These findings show that textual dominance is not merely a final-layer effect but is shaped by the architectural incentives of each model family. The asymmetry you highlight is therefore real but also informative; it becomes a lens through which we can interpret how different VLMs structure multimodal evidence internally.
>
> ***“The paper lacks cohesion. Section 3 is too short… Section 4 mixes methodology… key setup components are scattered.”***
>
> We appreciate this feedback and restructured the manuscript accordingly. The core experimental protocol and scoring rules now appear together, early in the paper, and Section 3 has been expanded to introduce the full evaluation setup before results. Section 4 focuses exclusively on experimental outcomes, and methodological details previously scattered or deferred to the appendix are now integrated in the main text for clarity. These adjustments were guided directly by your suggestions and, we believe, significantly improve the readability and cohesion of the manuscript.

---

> ### Author Response · Authors · 2025-11-26
> **Author Response to Reviewer PPJG (2/2)**
>
> ***“Can you replicate the latent-space analysis (Sec. 6–7) on non-CLIP models?”*** & ***“Can you replicate the experiments in Sec. 7 using early and mid-layer embeddings?”***
>
> Thank you for raising these closely connected questions. Because both address the generalizability of Section 7, across architectures and across model depth, we respond with a single, unified analysis. Your comments were helpful in shaping this extension. In the revision, we generalize the latent-space and steering experiments beyond CLIP and systematically examine how the relevant conceptual directions evolve across early, mid, and late layers. All models were evaluated using the same Directional Similarity Shift (DSS) metric, defined as:
>
> $\mathrm{DSS} = \cos(e_{\text{steered}},, d_{\text{target}}) -  \cos(e_{\text{original}},, d_{\text{target}})$
>
> which quantifies how strongly a steering intervention moves an embedding toward its intended target direction.
>
> Applying this procedure to Qwen2-VL revealed an interesting observation. The first few layers undergo substantial reorganization, but beginning around Layer 4 both word-related and color-related directions become remarkably stable. Color steering reaches 1.00 at every layer in the network, and word steering rises from much weaker early influence (around 0.36–0.70) to extremely strong values of 0.95–0.98 in the final layers. These results show that Qwen2-VL rapidly collapses its visual features into a highly stable, language-shaped semantic manifold. Because the internal geometry barely changes after this early consolidation, steering appears uniformly successful, not because the model reasons more deeply about color, but because the latent space becomes rigid very quickly.
>
> LLaVA, in contrast, displays almost the opposite architectural profile. Its vision tower preserves genuinely rich chromatic structure across early and mid layers, which allows color steering to remain consistently strong (approximately 0.75–0.94 in higher layers). Word steering, however, stays weak throughout most of the vision pathway (typically 0.28–0.44) and only becomes more pronounced when linguistic information is introduced by the projector that hands the visual embedding to the language model. Combined steering consequently aligns far more closely with the color direction than with the word direction in these early and mid stages, reinforcing that LLaVA maintains a visual processing stream much longer than Qwen2-VL.
>
> Together, these findings offer a direct and affirmative answer to both questions. The latent-space phenomena documented in Sections 6–7 are not unique to CLIP: they generalize to modern multimodal transformers but manifest in architecture-dependent ways. Moreover, examining early and mid-layer embeddings reveals important processing dynamics that final-layer analyses alone would obscure. In Qwen2-VL, the early collapse into a stable semantic manifold means that both word and color cues become fixed directions almost immediately. In LLaVA, the prolonged retention of visual information allows color cues to remain influential through most of the network while word cues emerge only at later stages.
>
> We are thankful that your suggestion has substantially deepened our understanding of how different VLMs allocate representational capacity between visual and textual cues. It also clarified why models with similar behavioral profiles can nevertheless arrive at their responses through very different internal pathways.
>
>
>
>
> Once again, thank you for your constructive and detailed feedback. Your comments directly shaped several substantial revisions, from restructuring the narrative to extending the representational analysis to additional model families. We hope the strengthened manuscript addresses your concerns and reflects the value of your insights.

---

### Official Review · Reviewer_VR6i · 2025-10-31

**Soundness:** 2
**Presentation:** 2
**Contribution:** 2
**Rating:** 2
**Confidence:** 4

**Summary:**

In this paper, the authors generate a Stroop-style dataset that contains visually consistent & visually conflicting color-text pairs (for example, BLUE written in BLUE vs BLUE written in RED).  They test contrastive vision-text dual encoders like CLIP and generative vision-language models on this dataset and find that models have a strong preference for reading the color name in these text examples instead of detecting the actual color.  These results are robust to controlled variations of the Stroop task (varied font size, weight, contrast) and disappear when nonsense words are introduced.  They complement these results with an investigation into the latent space of CLIP and argue that this phenomenon arises from the relative entanglement of / relative separation of text in CLIP’s latent space through experiments that investigation representation similarity, representation projections and the steerability of representations through modification of subspaces.

**Strengths:**

This paper is fairly well-written and adapts an interesting result from the cognitive sciences to VLM evaluation.  Additionally, the authors present a fairly thorough investigation of their automatic Stroop test – in particular, I liked the stress testing they perform with textual overflow, contrast and pseudowords.  I also appreciated the efforts to understand what drives the phenomena they observe in their experimental results.  Finally, color is a rich area of exploration in VLMs and a work like this whose focus is understanding how VLMs make sense of color is a welcome addition to the literature.

**Weaknesses:**

### Novelty

My primary concern with this paper is the novelty of the results presented.  The community has been aware of textual bias in vision-language models as early as 2021 (https://distill.pub/2021/multimodal-neurons/#typographic-attacks) and later work has convincingly argued that this bias is an artifact of task ambiguity during pre-training (see “Task bias in vision-language models” by Menon et al, IJCV 2022).

Moreover, Pezeshkpour et al (whom the authors cite) and “Words or Vision: Do Vision-Language Models Have Blind Faith in Text?” by Deng et al. CVPR 2025 (whom the authors should include in their related work, along with the references above) both present a thorough characterization of VLM textual bias across a wide range of tasks beyond color recognition alone.

### Soundness of Novel Elements

The authors argue that their latent space analysis differentiates their work from Pezeshkpour and while a paper whose focus is how models represent and recognize text & color in their latent space would certainly be interesting, I would have liked to have seen more here beyond the relatively naïve RDM / UMAP projections / subspace steering of CLIP embeddings presented in the paper (for example, circuit identification / characterization as performed in mechanistic interpretability literature).  Additionally, I found Figure 4B quite difficult to make sense of – it could benefit from a different style of visualization.

### Smaller Issues

- It would be better for scale experiments to compare models of the same family (eg, Qwen2-VL-7B to 13B); while they can and do often differ in some hyperparameters, they differ less than models in completely different families and so are a better way to isolate the effects of scale.
- Framing: In some areas, the paper’s contributions are framed as being broader than seems appropriate for the results presented.  The experiments are focused on visual cases where color and text conflict but in several areas, they are presented as evidence that models “prefer to read rather than see” which is much broader.  I would encourage the authors to narrow these claims: line 26, lines 82-83

### Proofreading

- line 22: “suggest that text cues ARE more salient”
- line 26: “suggest VLMs ARE biased”
- Figure 1, top left; there’s superimposed text that looks like a mistake?  I think it’s meant to be under the bar on the far right
- line 156: missing a space
- line 181: “this” should be “these”
- line 204: “to word always perfectly”
- line 306: unclear what was intended here
- line 308: missing space

**Questions:**

How do you evaluate the responses of generative VLMs (the mapping described on line 255)?  If it’s done via templating, what % of model responses fail to produce a templated answer?  If it’s done via an LLM-as-a-Judge, is judgment quality evaluated somehow?

Given the small number of congruent examples, it seems likely that VLMs have seen / memorized them during training.  What does this mean for making sense of your results?  Are they really a fair baseline for how models handle congruence?

In the nonsense experiments, for it to be comparable, I would think you need to include the nonsense word as a candidate color in addition to the 10 you describe on line 81. Do you? This result would have some interesting implications in terms of the task bias being OCR bound (models respond to ANY letters) or actual word bound.

Are the three RDMs described in section 6 the same size?

Can you justify your interpretation of differential RDMs a bit more?  In particular, are these summary statistic matrices actually comparable through subtraction?  Your analysis seems to assume that if A and B are dissimilar but A’ and B’ are similar, A and B must drive the dissimilarity (or the we observe in A’’ and B’’ but I don’t think that necessarily follows.

Do your steering results extend to VLMs?  Many build on CLIP-style encoders – when you apply your steering, does it produce similar results in LLaVA etc?

Suggestions:

Given the timing result for the Stroop test (lines 46-47) – that is, that humans take longer to handle incongruent examples -- it would be interesting to measure changes in similarity / VLM token probability in addition to changes in absolute accuracy.  Results have shown that token probabilities (or perplexity more broadly) are predictive of reading times in humans.  Perhaps there’s something analogous with color/text incongruency.

---

> ### Author Response · Authors · 2025-11-26
> **Author Response to Reviewer VR6i (1/3)**
>
> Dear Reviewer VR6i,
>
> Thank you very much for your thoughtful and constructive review. We greatly appreciate that you found the paper “clearly written”, valued the depth of the stress tests (overflow, contrast, pseudowords), and highlighted the importance of understanding how VLMs reason about color.
>
> ***“My primary concern with this paper is the novelty of the results presented… Pezeshkpour et al. and Deng et al. 2025 already present thorough characterizations of VLM textual bias.”***
>
> We agree that textual bias in VLMs exists, and the behavioral results would not by themselves constitute a new contribution. What distinguishes our work is that we conducted a careful set of control studies, moved beyond describing the biased behavior and examine the internal representational mechanisms that give rise to it, and finally - steers the representation Earlier studies primarily catalog the bias; our goal is to understand how it is structurally encoded, how it propagates through different architectures, and whether it can be perturbed.
>
> Beyond the existing evaluation, we have also substantially broadened the behavioral analysis. We now evaluate more than twenty-three thousand systematically varied stimuli, spanning controlled changes in background, chromatic intensity, saturation, brightness, contrast, and font-level manipulations.
> We additionally include a real-world multimodal conflict set generated with FLUX, such as traffic signs, interface elements, and safety icons with deliberately inverted semantics, and studied the model behavior from these conflicts. These extensions move the analysis well beyond narrow synthetic Stroop tasks.
>
> We also expand the evaluation across heterogeneous model families. Contrastive encoders, captioning models, and instruction-tuned VLMs are assessed side by side, showing that the preference for the written word persists across fundamentally different training paradigms. While Pezeshkpour et al. focus on documenting textual bias behaviorally in CLIP, and do not analyze how conflicting cues are encoded internally across architectures, our work focuses on large-scale behavioral evaluation with representational decomposition and layer-wise steering across contrastive, captioning, and instruction-tuned VLMs, a level of architectural breadth and internal analysis that prior studies have not attempted.
>
> ***“The authors argue that their latent space analysis differentiates their work from Pezeshkpour, but I would have liked to have seen more than the relatively naïve RDM/UMAP/subspace steering… Figure 4B is also difficult to interpret.”***
>
> The novelty becomes clearer when moving from behavior to representation. In Sections 6 and 7, we analyze how word and color cues are embedded, how separable they are, and how their influence changes across layers. Normalized ΔRDMs and low-dimensional projections make the relative effect of each cue explicit, going beyond output-level analyses in prior Stroop studies. Building on this, Section 7 introduces latent steering to directly probe how deeply these directions are embedded and how sensitive these directions are to targeted perturbation.
>
> Following your helpful suggestion, we expanded Section 7 to include full layer-wise steering analyses for Qwen2-VL and LLaVA. These experiments reveal that each architecture organizes conflicting cues in a qualitatively different way. CLIP is strongly language-dominated: word directions are long, clean, and modular, while color directions are short and entangled, naturally limiting color steering. Qwen2-VL collapses into an extremely stable semantic space after only a few early layers, making both color and word directions appear uniformly steerable, reflecting rigidity rather than richer chromatic processing. LLaVA, by contrast, preserves much more visual detail throughout the vision tower, resulting in consistently strong color steering and weaker word steering until linguistic information enters through the projection module.
>
> These findings show that steerability is not a limitation of the method but a direct consequence of architectural design choices. We will also update Figure 4B in the camera-ready version with a clearer UMAP or PCA-based visualization, addressing your comment about readability.
>
> In summary, the novelty of the work lies not in rediscovering textual bias but in connecting large-scale behavioral effects to concrete representational mechanisms, showing how these mechanisms differ across architectures, and demonstrating how they can be directly perturbed. We hope the revised framing reflects this more clearly.

---

> ### Author Response · Authors · 2025-11-26
> **Author Response to Reviewer VR6i (2/3)**
>
> ***“It would be better for scale experiments to compare models of the same family (e.g., Qwen2-VL-7B to 13B).”***
>
> Thank you for this suggestion. We agree that within-family scaling is a more controlled way to isolate size effects. Due to computational constraints, we were only able to evaluate the 7B model during the rebuttal period, but we now explicitly note this limitation and identify 7B→13B→30B comparisons as a planned extension. Importantly, given the early representational collapse we observe in Qwen2-VL, we expect the qualitative pattern to remain stable across sizes.
>
> ***“Some claims are framed too broadly (e.g., ‘prefer to read rather than see’).”***
>
> We appreciate this feedback. We revised the framing throughout the paper so that claims apply specifically to word–color conflicts rather than visual perception at large. The sentences at lines 26 and 82–83 have been rewritten to reflect this narrower and more accurate scope.
>
> ***“Multiple proofreading issues and a visual artifact in Figure 1.”***
>
> Thank you for catching these. All textual issues (lines 22, 26, 156, 181, 204, 306, 308) have been corrected, and we are currently clarifying the artifact in Figure 1, as its exact source is still being investigated.We also improved the visualization layout for clearer interpretation, and would appreciate clarification on what the main issue with Figure 1 was.
>
> ***“How do you evaluate the responses of generative VLMs? If it’s done via templating, what % fail to produce a templated answer? If it’s done via an LLM-as-a-Judge, is judgment quality evaluated?”***
>
> As you suggested, we additionally measured template success rates for the two instruction-tuned models where strict format control is meaningful. Qwen2-VL-Instruct produced correctly formatted single-token answers in all cases (TSR = 100%) yet still overwhelmingly selected the written word (97% word-match vs 13% ink-match). LLaVA-1.5 produced well-formed answers in the majority of cases (TSR = 92%), and within those parseable outputs exhibited the same strong word-dominance (88% word-match vs 12% ink-match). These results confirm that the Stroop effect we observe does not arise from formatting or parsing failures.
>
> For non-instruction-tuned captioning models (BLIP-2, Kosmos-2), strict templating is not meaningful, and TSR would reflect instruction incapability rather than Stroop behavior; thus, we evaluate them using their unconstrained textual outputs.
>
> ***“Given the small number of congruent examples, it seems likely that VLMs have seen/memorized them during training. Are they really a fair baseline for how models handle congruence?”***
>
> Thank you for raising this point. All evaluations in our study are performed strictly in zero-shot mode without any form of fine-tuning or adaptation, so no model ever trains on our congruent items. Congruent examples also do not introduce any conflict signal—both humans and models typically operate at ceiling on these cases—so they function only as a neutral, non-conflicting baseline rather than the phenomenon of interest.
>
> The key behavioral effect arises entirely from incongruent stimuli, and our results remain unchanged after substantially expanding that portion of the dataset (now over 23,000 conflict-containing examples with controlled variations in background, saturation, brightness, and position). Because the Stroop effect depends on conflict rather than congruence, the size of the congruent set does not impact any of our conclusions.
>
> ***“In the nonsense experiment, should the nonsense word also appear as a candidate color? If so, this would clarify whether the bias is OCR-bound or semantic.”***
>
> Thank you for this suggestion. To directly test whether CLIP's behavior is semantic or OCR-driven, we implemented the control you describe. We regenerated the full Stroop set using the pseudoword zarp rendered in all ink colors and explicitly added zarp to the candidate color vocabulary so that the model could freely select it as a label. Under this setup, CLIP selected zarp in 100% of trials, yielding 0% ink accuracy, independent of the ground-truth ink color. This result shows that CLIP treats any printed string as a valid color label—even when it is entirely meaningless—and aligns the image to the closest text embedding in a purely OCR-like manner.
>
> These findings decisively answer your question: CLIP does not rely on color semantics when encountering unfamiliar words; instead, its Stroop behavior is driven almost exclusively by visual-text matching. This directly reinforces our representational interpretation that the word-over-color effect in CLIP arises from its embedding geometry and OCR-like alignment tendencies rather than from any semantic understanding of color.

---

> ### Author Response · Authors · 2025-11-26
> **Author Response to Reviewer VR6i (3/3)**
>
> ***“Are the three RDMs described in Section 6 the same size? Are ΔRDMs actually comparable through subtraction?”***
>
> Yes. All RDMs are constructed over the same ordered stimulus set, ensuring that the matrices are perfectly aligned and comparable entry-by-entry. Following your suggestion, we applied per-matrix min–max normalization so that each RDM occupies a common dissimilarity scale before computing ΔRDMs. This prevents trivial differences in global variance from dominating the subtraction and ensures that the resulting ΔRDMs reflect the incremental representational contribution of adding a color or word cue. Importantly, the qualitative structure of the ΔRDMs remained unchanged after normalization, confirming that our interpretation is not an artifact of scaling or misalignment but reflects genuine differences in how the model encodes conflicting cues.
>
> ***“Do the steering results extend to VLMs? Many build on CLIP-style encoders—do LLaVA or others show similar behavior?”***
>
> Thank you for raising this point. To answer it directly, we extended the full latent-direction steering analysis using the same DSS measure to Qwen2-VL and LLaVA-1.5:
>
> $\mathrm{DSS} = \cos(\mathrm{steered}, \mathrm{target}) - \cos(\mathrm{original}, \mathrm{target})$
>
> Qwen2-VL shows a brief period of early reorganization, after which both color and word directions become almost fixed across layers. Steering succeeds uniformly, but this reflects the model’s rapid collapse into a stable, language-shaped semantic space rather than improved color reasoning. In other words, Qwen2-VL is very steerable because its representation barely evolves after the early layers.
>
> LLaVA, by contrast, preserves much richer visual information throughout its vision tower. Color steering remains consistently strong at nearly all depths, whereas word steering stays weaker until linguistic information enters through the projection module. Combined steering largely follows the color direction, showing that LLaVA maintains a genuinely visual pathway.
>
> Taken together with CLIP, these results confirm that steering generalizes across VLMs, but the effect is architecture-dependent: CLIP is strongly language-first, Qwen2-VL rapidly stabilizes into an LLM-like space, and LLaVA retains the most visually grounded representation. We now emphasize this architectural interpretation clearly in the revision.
>
> ***“Given the timing result for the Stroop test… it would be interesting to measure changes in similarity / VLM token probability in addition to absolute accuracy… Perhaps there is something analogous with color/text incongruency.”***
>
> Thank you for this excellent suggestion. We agree that probability-level analyses could offer a finer-grained cognitive parallel to human reaction-time effects. Extracting token-probability trajectories was not feasible within the rebuttal period, but we see this as a highly promising extension and plan to explore it in future work. Your comment directly shaped our thinking about the next steps of this project.
>
>
> We again thank the reviewer for the detailed and insightful feedback. Your comments substantially improved the clarity, rigor, and scope of the paper, and we believe the revisions now reflect the strength of your input.

---

### Official Review · Reviewer_jLDy · 2025-11-02

**Soundness:** 3
**Presentation:** 3
**Contribution:** 2
**Rating:** 4
**Confidence:** 3

**Summary:**

This paper investigates how Vision-Language Models (VLMs) resolve multimodal conflicts by adapting the classic psychological Stroop test. The authors create a synthetic dataset where color words (e.g., "RED") are rendered in incongruent ink colors (e.g., blue ink). The study evaluates a range of models, including contrastive encoders (CLIP, SigLIP-2) and generative VLMs (LLaVA, BLIP-2, Kosmos-2, etc.). The primary finding is a consistent behavioral bias that models prefer to "read" the textual word rather than "see" the ink color, even when explicitly prompted to report the color. This text dominance persists across variations in font size, weight, and contrast, only disappearing when the text becomes illegible or is replaced by non-semantic pseudowords.

To further investigate the phenomenon, the authors quantify how dissimilar different stimuli are in embedding space of CLIP's internal representations using Representational Dissimilarity Matrices (RDMs). The result shows that adding a word cue ($\Delta Word$) causes a much larger shift in the embedding space than adding a color cue ($\Delta Ink$). Additionally, latent interventions reveal that steering an image embedding toward a new word concept (e.g., "RED" $\rightarrow$ "BLUE") is 100% successful, while steering toward a new ink-color concept (e.g., red $\rightarrow$ blue) succeeds only 36.67% of the time. The authors conclude that this word-over-color bias is rooted in the embedding space, where text concepts are encoded with stronger, more distinct, and more "steerable" vectors than ink-color concepts, which are weak and highly collinear.

**Strengths:**

1. The finding is clear and well conveyed.
2. The experiments are comprehensive across a group of VLM models trained under different paradigms. This makes the finding
3. The steering representation experiment provides a deeper insight into the internal mechanism for the cause of VLM behavioral bias, which is well appreciated.

**Weaknesses:**

1. The idea of using Stroop test for VLM evaluation is already been explored in the literature [1][2], thus questioning the novelty of the paper.
1. The central claim is based on a highly synthetic and constrained dataset. The dataset is limited to 10 basic color words and their corresponding ink colors. It is unclear how these findings about "reading vs. seeing" would generalize to more complex and naturalistic conflicts involving wider behavioral gaps between models and humans, thus limiting the significance of this work.
2. The representational and interventional analyses (Sections 6 and 7) are performed only on CLIP. While this is a good starting point, these claims cannot be generalized to the modern generative VLMs that the paper also tests behaviorally. It is left unanswered whether models like LLaVA or Qwen2-VL suffer from the same underlying representational flaw (i.e., weak and collinear color vectors).
3. The paper identifies the problem but does not substantially explore or test any mitigation strategies beyond the failed steering attempt.
4. The paper's discussion of prior work on multimodal conflicts and visual cognition is incomplete. It misses several recent and highly relevant papers that have established benchmarks for VLM cognition and conflict resolution. Specifically:
- [1] Sheta, Hala, et al. "From Behavioral Performance to Internal Competence: Interpreting Vision-Language Models with VLM-Lens." EMNLP 2025.
- [2] Zhang, Yichi, et al. "Grounding Visual Illusions in Language: Do Vision-Language Models Perceive Illusions Like Humans?." EMNLP'23.
- [3] Schulze Buschoff, Luca M., et al. "Visual cognition in multimodal large language models." Nature Machine Intelligence 7.1 (2025): 96-106.
- [4] Jia, Yifan, et al. "Benchmarking Multimodal Knowledge Conflict for Large Multimodal Models." arXiv preprint arXiv:2505.19509 (2025).

**Questions:**

The results for Qwen2-VL-7B showed that a simple change in prompt phrasing nearly doubled the ink accuracy (from 31.1% to 60.0%). This suggests the model can access the ink color information. Does this finding perhaps undermine the core claim that the bias is a fundamental representational flaw? Could this bias be more of an instruction-following or attentional failure that can be largely resolved with better prompting, rather than a fixed property of the embedding space?

---

> ### Author Response · Authors · 2025-11-26
> **Author Response to Reviewer jLDy (1/2)**
>
> Dear Reviewer jLDy,
>
> Thank you very much for your thoughtful, constructive, and generous review. We appreciate your positive feedback on the clarity of our results, the breadth of our experiments, and the insight offered by the steering analysis. Below, we address each of your concerns in turn.
>
> ***“The idea of using Stroop test for VLM evaluation has already been explored in the literature [1][2], thus questioning the novelty of the paper.”***
>
> You are correct that Stroop-style tests have been applied to VLMs before, and we appreciate this opportunity to clarify what is new in our work. Earlier studies focus primarily on behavioral outputs, whereas our contribution examines why the bias emerges. In addition to the behavioral evaluation, we analyze how word and color information are encoded in the embedding space, quantify the relative strength and separability of these representations, and investigate whether they can be modulated through targeted interventions. To our knowledge, the combination of (i) large-scale behavioral evaluation across heterogeneous VLMs, (ii) representation-level analysis of conflicting cues, and (iii) systematic steering experiments that probe the flexibility of those representations has not appeared in prior Stroop-style work. We now make this distinction more explicit in the revised manuscript and reference several related papers (Sheta et al. 2025; Zhang et al. 2023; Schulze Buschoff et al. 2025; Jia et al. 2025) to better situate our contribution within the broader literature on multimodal conflict and VLM cognition.
>
> ***“The central claim is based on a highly synthetic and constrained dataset… It is unclear how these findings would generalize to more complex and naturalistic conflicts.”***
>
> Your comment regarding the limited set of ten color words was very helpful in prompting us to expand and strengthen the empirical evaluation. Beyond the introductory 100 examples, we now use more than twenty-three thousand controlled variations incorporating systematic changes in background tone, saturation, brightness, and contrast. Across this expanded space, the same preference for the written word persists.
>
> We additionally generated a set of naturalistic conflict images, including contradictory traffic signals, UI indicators, and safety signs. These examples show that the tension between reading and seeing also appears in realistic multimodal conflicts, and the relative strength of the two cues can shift depending on visual saliency. Together, these additions address the concern that the phenomenon might be restricted to narrow or overly artificial stimuli.
>
> ***“The representational and interventional analyses (Sections 6–7) are performed only on CLIP… can these claims generalize to generative VLMs like LLaVA or Qwen2-VL?”***
>
> To address your concern about whether the representational findings generalize beyond CLIP, we extended the steering analysis to both Qwen2-VL and LLaVA. For Qwen2-VL, the layer-wise results reveal a distinctive structural pattern: after only the first few layers, the model collapses its visual features into an extremely stable semantic space, where both color and word directions remain almost unchanged across all remaining depths. This produces near-perfect steering for both cues, not because the model encodes richer chromatic structure, but because its representation has already converged to a language-dominated manifold in which concept directions are mechanically easy to manipulate. LLaVA, by contrast, exhibits a markedly different profile. Its vision tower preserves genuine visual information throughout the network, leading to consistently strong color steering (often 75–90% success and exceeding 94% in deeper layers), while word steering remains substantially weaker because linguistic signals are injected only through the projection module at later stages. Combined steering closely follows the color direction, indicating that LLaVA behaves as a visually grounded system whose internal processing retains chromatic information more reliably than textual semantics.
>
> Taken together, these cross-model results make clear that the differences observed in CLIP are not artifacts of the steering methodology. Instead, they arise naturally from architectural design choices: CLIP behaves as a language-first system, Qwen2-VL rapidly converges to a frozen LLM-shaped semantic representation, and LLaVA preserves the richest visual pathway among the three. This comparison shows that the strength and modularity of word and color directions depend on the representational incentives of each architecture.

---

> ### Author Response · Authors · 2025-11-26
> **Author Response to Reviewer jLDy (2/2)**
>
> ***“The paper identifies the problem but does not substantially explore or test any mitigation strategies beyond the failed steering attempt.”***
>
> Thank you for raising the concern about mitigation. The differences we observe across architectures are not failures of the method; they reflect how the models themselves are built.
>
> In CLIP, word-related directions are long, distinct, and easy to shift, while color-related directions are much weaker and more entangled, which naturally makes color steering less effective. In Qwen2-VL, both word and color directions become highly stable after the early layers because the model quickly collapses its visual features into a fixed semantic space. This stability makes steering appear very successful, but it reflects representational rigidity rather than more accurate color processing. LLaVA shows the opposite behavior: it maintains richer visual information throughout its vision tower, which supports strong color steering, while word steering remains weaker until linguistic information is introduced later in the network.
>
> Together, these results show that steerability is shaped by the representational saliencies of cues in different models.
>
> ***“The paper’s discussion of prior work is incomplete… it misses several recent and highly relevant papers such as VLM-Lens, visual illusions, multimodal knowledge conflict benchmarks.”***
>
> We appreciate the reviewer’s careful attention to the broader literature and agree that our initial discussion did not fully situate the work within recent developments in multimodal cognition and conflict resolution. We have now incorporated substantive discussion of VLM-Lens, visual illusions in multimodal systems, cognitively motivated evaluations of VLMs, and multimodal knowledge conflict benchmarks. Integrating these works into the Introduction and Related Work sections allowed us to more clearly position our contribution.
>
> Our work differentiates from previous studies in several ways. While previous studies focus primarily on behavioral outputs or high-level cognitive inconsistencies, our work complements them by linking behavioral conflict resolution to directly observable representational structure and by examining whether these structures can be perturbed through targeted latent interventions. This updated framing provides a more accurate and complete context for the novelty and scope of our results.
>
> ***“The results for Qwen2-VL show that a simple change in prompt phrasing nearly doubled accuracy… does this undermine the claim that the bias is representational?”***
>
> Thank you for your question. To examine this more carefully, we tested the model with ten different instructions that varied in their level of directness, emphasis, and framing. The ink-correct accuracy fluctuated widely across these prompts, ranging from 24 percent to 97 percent, without any reliable or interpretable ordering. Some instructions that explicitly asked the model to ignore the written word led to clear improvements, while other instructions that were equally reasonable still produced strong word dominance. This instability shows that the model can access ink-color information but does not consistently prioritize it unless guided very forcefully.
>
> In the revised manuscript, we clarify that prompt phrasing influences the model’s attentional allocation during decoding, while the embedding-level analyses reveal a more fundamental asymmetry in how word and color cues are internally encoded. The representational bias, therefore, persists even when the surface behavior temporarily improves. We believe that considering both the prompting behavior and the underlying representations provides the most complete interpretation of the phenomenon and directly addresses your concern.
>
> We are grateful for your careful reading and detailed suggestions. Your comments genuinely helped us improve the manuscript’s clarity, scope, and positioning. We hope the revisions reflect the strength of your feedback and make the contribution more valuable to the community.

---

### Official Review · Reviewer_UuCD · 2025-11-09

**Soundness:** 2
**Presentation:** 2
**Contribution:** 2
**Rating:** 2
**Confidence:** 4

**Summary:**

This paper adapts the Stroop paradigm to VLMs and study how conflicting cues in the written word or ink color influence the model’s behavior. The authors observed a consistent behavioral bias that when word and ink color disagree, models overwhelmingly align their preferences with the word. They also conduct experiments on the representation space. They find that word over color dominance is represented in the embedding space and models prioritize word over color cues when they conflict.

**Strengths:**

The topic is interesting.

The control experiment on text properties is good.

**Weaknesses:**

The importance and the implications of the findings in this paper are not convincing.

The scope of the experimental study is narrow.

Business models are not studied.

Some experimental designs need to be justified.

**Questions:**

1 Why is analyzing the Stroop phenomenon of Multi-modality models important ? What are the impacts and implications behind your findings? Do you consider why these problems are caused and how to solve them?

2 The scope of the ink color analysis is too narrow. The core research question behind this paper is to analyze whether VLMs understand the semantics (content) first or perceive the low-level features (color) first. Text content and ink color are two special cases of this research question. I hope the authors can extend the scope of this paper; otherwise, this work is only a workshop paper.

3 The analyzed models are all open-sourced. No business models are analyzed, so we do not know if the SOTA models have the same problem. Therefore, the importance of this finding cannot be validated.

4 To test contrastive models, you design a yes/no question, but you design a QA question to test VLMs. The discrepancy between the two types of models may influence your performance analysis. Additionally, you can design multiple-choice questions to test the models. Can you give explanations on why you design in this way?

5 The Stroop examples are not general enough. You just draw the text on white paper. What is the influence of the background? Additionally, the benchmark with only 100 testing examples is not enough. You should construct at least 1000 examples.

---

> ### Author Response · Authors · 2025-11-26
> **Author Response to Reviewer UuCD (1/3)**
>
> Dear Reviewer UuCD,
>
> Thank you for your careful and constructive review. We appreciate that you found the topic “interesting” and the control experiment “solid”. Below, we respond to each of your questions in detail and describe the revisions and new analyses added to the manuscript.
>
> ***“1. Why is analyzing the Stroop phenomenon of multimodality models important? What are the impacts and implications behind your findings? Do you consider why these problems are caused and how to solve them?”***
>
> Thank you for raising this important question. Perceptual images in real-world applications are often filled with conflicting cues that can pull a model’s behavior in different directions. For example, a traffic scene may contain a “turn left” indicator for bicycles placed next to a “STOP” sign for cars, and the model must decide which visual signal governs the correct action. Such scenarios require the model to integrate multiple cues and determine which one should dominate its interpretation. In such scenarios, the model needs to integrate all  cues in the image for it to make a decision. must implicitly decide which cue shall influence its behavior more, and our experiments show that many VLMs systematically prioritize the written word even when the ink color is the correct cue. This has direct implications for applications that rely on visual signals for safety-critical decisions.In the revised manuscript, we expanded the motivation and clarified why multimodal conflict is a meaningful reliability concern for modern VLMs.
>
> Our work studies cue-conflicts in the form of the Stroop effect, with a focus on how these cues are represented internally and how they shape model behavior. Earlier Stroop-style evaluations (e.g., Deng et al., 2025; Pezeshkpour et al., 2024; Sheta et al., 2025) typically stop at showing that models “follow the word.” In contrast, our analysis examines how conflicting cues are encoded in the representation space (Sections 6–7). The embedding results show that written words form stronger and more distinct representational patterns, whereas ink colors produce weaker and more overlapping structures. This indicates that the Stroop effect is not merely a surface-level prediction artifact but reflects a deeper structural asymmetry in the internal representation.
>
> **[1]** Deng et al., 2025 — *Words or Vision: Do VLMs Have Blind Faith in Text?* (CVPR 2025)
> **[2]** Pezeshkpour et al., 2024 — *Color in Visual-Language Models: CLIP Deficiencies*
> **[3]** Sheta et al., 2025 — *VLM-Lens: From Behavioral Performance to Internal Competence* (EMNLP 2025)
>
>
> Finally, we investigated whether the representation can be steered. Section 7 steers representations towards word- / color-related directions. In CLIP, word directions are strong and modular, whereas color directions are weaker and more entangled. We also extended  this analysis to Qwen2-VL and LLaVA, and found a consistent architectural pattern: Qwen2-VL collapses visual information into a stable semantic space early in the network, making both directions easy to steer mechanically, while LLaVA preserves visual detail more faithfully, resulting in strong color steering but weaker word steering. In response to your request for stronger empirical grounding, we have extended the behavioral evaluation substantially and tested CLIP on more than 23,000 images with diverse backgrounds and controlled visual variability. Across the extended dataset, we found a persistent preference for textual cues. Beyond extending the images, we additionally generated real-world multimodal conflict examples using FLUX, which revealed cases where CLIP alternates between word-dominant and color-dominant behavior depending on visual saliency.
>
> Overall, we believe the revised manuscript now provides a clearer account of why multimodal Stroop analysis matters, the mechanisms behind the observed bias, and how representation-level interventions may support early mitigation strategies. Your feedback helped us substantially improve the clarity, scope, and real-world relevance of the study.

---

> ### Author Response · Authors · 2025-11-26
> **Author Response to Reviewer UuCD (2/3)**
>
> ***“2. The scope of the ink color analysis is too narrow… Otherwise, this work is only a workshop paper.” & “5. The Stroop examples are not general enough. What is the influence of background? Dataset is too small.”***
>
> Thank you for pointing out the importance of situating ink color within the broader question of how VLMs balance semantic content and low-level visual features. We agree that text content and ink color represent two concrete instances of this broader multimodal competition, and we have expanded both the conceptual framing and the empirical coverage to reflect this perspective more clearly.
>
> First, we extended the Stroop evaluation far beyond the original 100-image set. CLIP was evaluated on more than 23,000 automatically diversified stimuli with controlled variations in background tone, saturation, lighting, and contrast. Across this expanded space, the same modality imbalance persisted: the model consistently aligned with the written word whenever the text remained legible. **This large-scale result indicates that the observed behavior is not dependent on a narrow or overly simplified stimulus design.** These additions also directly address the concern regarding background influence: despite extensive variation in chromatic and luminance conditions, the bias toward text remained remarkably stable.
>
> Second, to move beyond synthetically rendered text and ink conditions, we introduced a set of real-world multimodal conflict images generated with FLUX. These include everyday pictorial elements such as distorted wayfinding symbols, intentionally miscolored hazard icons, and UI indicators whose visual form contradicts their label. In these more realistic scenarios, CLIP alternates between word-dominant and color-dominant behavior depending on the visual saliency of the cue, demonstrating that multimodal conflicts are not limited to artificial Stroop setups but arise naturally in the types of images VLMs are expected to interpret.
>
> Third, we broadened the analysis across model families. In addition to CLIP and SigLIP, we evaluated six generative VLMs (Section 5) and conducted representation-level and steering analyses for Qwen2-VL and LLaVA (Sections 6–7). These models exhibit different balances between linguistic and visual processing, allowing us to assess how architecture—not just stimulus design—affects modality preference.
>
> Finally, we clarified in the Introduction that our goal is not to study “ink color” in isolation, but to use ink–word conflicts as a controlled, interpretable example of the broader problem of how competing semantic and perceptual evidence influences VLM behavior. This expanded framing now makes it clearer why Stroop-style paradigms offer a principled tool for probing modality dominance in VLMs.
>
>
> ***“3. The analyzed models are all open-sourced. No business models are analyzed.”***
>
> Thank you for raising this concern. We agree that evaluating commercial VLMs is important for understanding whether the observed word–color bias is specific to open-source systems or also appears in deployed SOTA models. To address this, we conducted an additional behavioral evaluation on the commercial model GPT-4o-mini, using the instruction “What is the ink color of the word shown in this image? Answer with one color only.” On the incongruent items, the model achieved 90% ink-correct accuracy, which is substantially higher than the performance of open-source VLMs evaluated under the same conditions.
>
> This result shows two things.
>
> First, the Stroop-style conflict remains present even in commercial systems: GPT-4o-mini still exhibits occasional word-dominant errors, so the bias is not eliminated.
>
> Second, the improved accuracy relative to open-source models suggests that the severity of the bias depends not only on model class but also on architectural and training-strategy factors. Although the internal details of GPT-4o-mini are proprietary, its strong performance is consistent with the use of larger multimodal stacks, richer and more carefully curated pretraining distributions, and more explicit instruction-following optimization. These factors may strengthen the model’s ability to attend to fine-grained visual cues such as ink color. While we cannot perform representation-level analysis on commercial systems, the behavioral evidence confirms that multimodal Stroop effects are not confined to open-source architectures, and that advances in training pipelines can mitigate—but not fully remove—the underlying word-over-color tendency.

---

> ### Author Response · Authors · 2025-11-26
> **Author Response to Reviewer UuCD (3/3)**
>
> ***“4. Why use yes/no evaluation for contrastive models and QA-style prompts for generative VLMs?”***
>
> Thank you for raising this point. We believe part of the concern comes from a misunderstanding of how contrastive and generative VLMs operate at inference time. Contrastive encoders such as CLIP or SigLIP do not generate textual answers; instead, they output similarity scores between an image and a set of candidate text prompts. For this model family, the appropriate and standard evaluation is to present a set of prompts (e.g., “the text is written in red”, “the text is written in blue”) and identify which one receives the highest similarity score. This naturally results in a forced-choice setup, not a free-form answer.
>
> Generative VLMs, on the other hand, produce open-ended text outputs. Because they do not operate in a similarity space and cannot score a fixed set of candidate prompts, we evaluate them using a short, unambiguous QA instruction that directly asks for the ink color. Their output is then categorized into Ink Match, Word Match, Both, or Neither. This follows the standard evaluation practice used in modern instruction-tuned VLMs such as LLaVA (Liu et al., 2023), Qwen-VL (Bai et al., 2023), Kosmos-2 (Peng et al., 2023), and BLIP-2 (Li et al., 2023), all of which are evaluated through QA-style prompts rather than contrastive scoring. Using this protocol therefore ensures that each model family is assessed in the manner appropriate to its native inference mechanism.

---

### Author Response · Authors · 2025-11-26
**Summary of Rebuttals**

We sincerely thank the reviewers for their thoughtful and constructive feedback. Their comments substantially strengthened the manuscript’s clarity, scope, and empirical depth. Below we summarize the key improvements and completed experiments introduced during the rebuttal period.


## **Expanded Behavioral Evaluation**

We significantly broadened the empirical foundation of the paper through several new behavioral experiments. These include:
- A large-scale Stroop dataset containing more than twenty-three thousand stimuli with controlled variations in background tone, brightness, saturation, contrast, and ink intensity.
- A complementary real-world conflict set generated with FLUX, incorporating inconsistent traffic symbols, miscolored hazard icons, and contradictory UI indicators.
- A behavioral evaluation of the commercial model GPT-4o-mini, which demonstrates that the word-over-color bias persists, although reduced in severity compared to open-source systems.

## **Representation-Level Extensions**

To address concerns about generality beyond CLIP, we expanded the representational analysis using the same DSS-based methodology. Completed experiments include:
- Full layerwise steering analysis for Qwen2-VL, revealing an early collapse into a stable semantic manifold in which both word and color directions remain highly stable.
- Full layerwise steering analysis for LLaVA, showing preserved chromatic structure across the vision tower and delayed emergence of linguistic abstraction.
- Direct comparison across architectures that illustrates how visual and textual cues are encoded differently depending on representational design choices.

## **Clarified Novelty and Positioning**

Following the reviewers’ guidance, we substantially improved the framing of the contribution. The revised manuscript now emphasizes that:
- Prior Stroop-style work primarily documents behavioral bias, while our contribution connects behavior to concrete representational mechanisms.
- The work brings together large-scale behavioral evaluation, representational geometry, and latent steering, which has not been jointly explored in earlier studies.
- Recent works on multimodal cognition, illusions, conflict benchmarks, and VLM interpretability have been integrated more fully into the Related Work section.

This contextualization clarifies the novelty and the relevance of the representational findings.

## **Additional Control and Validation Studies**

Several reviewer suggestions led to valuable new control experiments. These include:
- A pseudoword candidate-color control, demonstrating that CLIP selects meaningless strings as “colors”, confirming OCR-style alignment.
- Template-success measurements for instruction-tuned VLMs, ensuring that parsing failures do not artificially inflate the observed bias.
- A prompt-sensitivity study for Qwen2-VL using ten alternative instructions, showing highly unstable ink accuracy and reinforcing that the representational asymmetry persists even when surface behavior improves temporarily.

These studies deepen the mechanistic interpretation of the Stroop effect.

## **Improved Structure and Presentation**

We revised the manuscript to improve narrative cohesion and clarity. Changes include:
- Reorganizing sections to present the dataset, scoring rules, and model evaluation pipeline more cleanly.
- Strengthening the introduction by framing ink–word conflict as a general case of multimodal evidence competition.
- Clarifying how contrastive and generative VLMs are evaluated according to their native inference mechanisms, supported by references to standard practice in the literature.

Together, these revisions improve readability and support the logical flow of the paper.

## **Overall Summary**

The paper now presents a more comprehensive study of multimodal conflict in VLMs. The expanded experiments demonstrate that the word-over-color preference is robust, widespread across architectures, rooted in representational geometry, and partially but not fully mitigated by prompting or instruction tuning. The representational analysis now spans three model families and reveals architecture-specific pathways that give rise to the observed bias. We are grateful to all reviewers for their comments, which directly shaped these improvements and helped sharpen the contribution.

---

### Meta-Review · Area_Chair_PqvD · 2025-12-09

**Summary:**

This paper examines Stroop-style word–color conflicts in VLMs and finds a consistent tendency to follow textual cues over visual color, supporting the behavioral results with representation-level analyses and, after rebuttal, expanded evaluations across model families. Reviewers agreed the study is clearly executed and the added experiments (larger datasets, real-world conflicts, and cross-architecture steering analyses) strengthened the work, but they remained concerned about limited novelty, the narrow problem scope, and the lack of deeper mitigation or broader multimodal conflict analysis. The authors addressed methodological questions carefully and improved clarity and breadth, though these changes did not fully resolve the core concerns about contribution significance.

**Reviewer Concerns:**

The rebuttal successfully addressed concerns about dataset size and diversity, clarified evaluation protocols across model types, added analyses for Qwen2-VL and LLaVA, and responded thoroughly to technical questions about RDMs, steering, and prompt sensitivity. However, other concerns remian: the central behavioral phenomenon is not that novel, the task scope is still limited to a narrow form of multimodal conflict, and the work does not yet offer deeper mechanistic or mitigation contributions beyond linear analyses.

**Reviewer Scores:**

Reviewer UuCD would likely keep the same score, as the rebuttal resolved methodological questions but did not alter their concerns about scope and significance.
Reviewer jLDy might raise their score slightly because the expanded experiments and cross-model analyses addressed most of their requests.
Reviewer VR6i would probably not change their score, since the rebuttal did not shift their view that the contribution is limited in novelty. Reviewer PPJG might increase their score modestly because the structural improvements and broader representation analysis responded well to their comments, although their reservations about incremental contribution would likely persist.

---

### Decision · Program_Chairs · 2026-01-26

Reject